


# Optimal Precursors Identification for North Atlantic Oscillation using CESM and CNOP Method

Bin Mu[1], Jing Li[1], Shijin Yuan[1], Xiaodan Luo[1], and Guokun Dai[2]

[1]Department of Software Engineering, Tongji University, Shanghai, China
[2]Department of Atmospheric and Oceanic Sciences & Institute of Atmospheric Sciences, Fudan University, Shanghai, China

**Correspondence:** Shijin Yuan (yuanshijin2003@163.com)

**Abstract.** The North Atlantic Oscillation (NAO) is the most prominent atmospheric seesaw phenomenon in the North Hemisphere. It has a profound influence on the westerly wind strength and storm tracks in North Atlantic, which further affect the winter climate in Northern Hemisphere. Therefore, the identification for optimal precursor (OPR) of the NAO event is of research value and practical significance. In this paper, the conditional nonlinear optimal perturbation (CNOP) method, which

has been widely used in research on the OPR of climatic event, is adopted to explore which kind of initial perturbation is most likely to trigger the NAO anomaly pattern in the Community Earth System Model (CESM). Since the adjoint model of CESM has yet to be developed, this kind of problem cannot be solved using traditional strategies based on gradient information provided by the adjoint model. We utilize an adjoint-free algorithm to solve CNOP in such a high dimensional numerical model, and OPRs of the NAO can be successfully identified. The results reveal that OPRs obtained by CNOP can cause the basic state

to develop into typical dipole mode, and the nonlinear process plays an important role in the last stage of the prediction period. The algorithm adopted in this work can avoid falling into a local optimum and is accelerated with multiple parallel frameworks to enhance performance. The solution scheme can also be generalized to the OPR research of other climate events or other complex numerical models.

## 1 Introduction

The North Atlantic Oscillation (NAO) is the continuous phase-reversing fluctuation in the meridional direction of the sea level pressure (SLP) field in the North Atlantic, which is mainly related to the interannual variation of the pressure over Azores and Iceland. It represents large-scale alterations in the SLP differences between the subtropical and subpolar regions of the North Atlantic (Haylock et al., 2007). As the dominant mode of atmospheric circulation variability in the northern hemisphere, the NAO is the result of complex nonlinear interactions between many spatiotemporal scales (Önskog et al., 2018). Previous

research has shown that the NAO may be strictly linked to the anticyclonic (cyclonic) Rossby wave breaking (Franzke et al., 2004) and can be viewed as a process with an e-folding time scale of about two weeks (Feldstein, 2000). The NAO index (NAOI) is a quantified indicator of the NAO, and its classical definition is the difference between normalized SLP over Iceland and Azores (Andersson, 2002). In the past decade, the turbulence of the winter NAO has been quite extreme, and it has contributed significantly to the warm winter phenomenon throughout Europe, the cold weather in the Northwest Atlantic





(Hurrell, 1995), the dipole precipitation pattern over northwest Europe and northwest Africa (Wassenburg et al., 2016) and the surface temperature variation (Pokorná and Huth, 2015), etc.

The NAO can be regarded as a nonlinear initial value problem (Woollings et al., 2008), and its optimal precursor (OPR) is a kind of initial perturbation that is most likely to develop into NAO events (Mu et al., 2014). Since the initial condition has
a significant influence on the predictability of the decadal variability (Zhang et al., 2016), the OPR can help us to understand the dynamical processes of the NAO state transition. The physical mechanism for triggering the NAO event can be discovered by investigating the developing process of the OPR. Moreover, the sensitive areas determined by the spatial structure of the OPRs are beneficial to the intensive observations, thus improving the forecast accuracy of the NAO state transition. Hence, the research on OPR is of widespread scientific research value to study the physical mechanism and enhance forecast skill for the
NAO. Further, exploring and optimizing the method for solving the OPR also has essential meanings. Although the phase and amplitude of NAO are affected by numerous factors (Bucha, 2014; Yu and Lin, 2016; Hansen et al., 2017), the characteristics of the NAO in the atmospherical process can be captured by the nonlinear models (Luo et al., 2007). As a new generation of ocean-atmosphere coupled model, the Community Earth System Model (CESM) has presented the outstanding performance of NAO simulation, and the low-frequency variability of the NAO has also been well demonstrated (Nandini, 2017; Wang et al.,
15   2015).

The conditional nonlinear optimal perturbation (CNOP) is a mature method for solving OPRs. It describes the initial perturbation that causes the largest prediction error under a specific constraint condition at the prediction time. CNOP is appropriate for predictability studies of climate events with simulating nonlinear motions of oceans and atmospheres (Mu et al., 2003), and can be concluded as an optimization problem with constraints. CNOP approach was initially adopted to identify the OPRs
of ENSO (Duan et al., 2004) and was gradually applied in research on the onset of blocking events (Mu and Jiang, 2011), Kuroshio large meander (Zhang et al., 2017b), and Indian Ocean dipole events (Mu et al., 2017b). Recently, Jiang et al. (Jiang et al., 2013) explore the OPRs that trigger the NAO events using CNOP, demonstrating that the amplitude induced by the self-interaction of perturbations in the onset of the $NAO^-$ is stronger than that in the onset of the $NAO^+$. On this basis, Dai et al. (Dai et al., 2016) investigate the relationship between the OPR and optimally growing initial error (OGE) using CNOP.
It is indicated that the two types of OGEs and the OPRs corresponding to the two types of NAO events have similar structures, and both of them can develop into dipole NAO anomaly patterns. These studies have proved that CNOP is a useful tool to investigate the onset of the NAO event. In their studies, the T21L3 quasigeostrophic global spectral model, which is a simple three-level model designed by Marshall and Molteni (Marshall and Molteni, 1993), is applied under ideal conditions. Due to the feature of the T21L3 model, they selected geopotential height as the characterized variable, and potential vorticity is the input
variable. For solving CNOP, Jiang et al. (Jiang et al., 2013) and Dai et al. (Dai et al., 2016) all used spectral projected gradient 2 (SPG2) algorithm (Birgin et al., 2001). The SPG2 was designed to solve the minimum problem with restraints by determining the gradients of the cost function (Guo-Dong, 2009). Several approaches with the same type have also been adopted to calculate the CNOP in the early years (Duan and Mu, 2006; Wang et al., 2012; Bo et al., 2014), such as the sequential quadratic programming (SQP) algorithm (Büskens and Maurer, 2000) and the limited memory Broyden-Fletcher-Goldfarb-Shanno (L-





BFGS) algorithm (Liu and Nocedal, 1989), etc. Since these algorithms rely on gradient information, the corresponding adjoint model needs to be called to obtain the gradients of the initial condition in the solving process.

However, these adjoint-based methods would have a high probability to produce local CNOPs when the objective function has multiple extreme values and would fail with large initial disturbance or long prediction time due to the strong nonlinearity

of the dynamical model. Crucially, these traditional adjoint-based algorithms are not feasible to solve CNOP in complicated operational models that do not have an adjoint available (Wang, 2010). In recent years, swarm intelligence algorithms are gradually applied in the research of the CNOP (Zheng et al., 2012; Yang et al., 2017). These algorithms determine the search direction from the position and fitness values of particles instead of gradients so that they can be extended to models without the adjoint model. It is also indicated that the swarm intelligence method still achieves global CNOP and has a shorter run

time in the situation of larger initial perturbations, longer prediction times, multiple extrema values (Zheng et al., 2017) and discontinuous objective functions (Mu et al., 2015). Although these algorithms have shortened the runtime, it is still very time-consuming to search CNOP in the original dimensions. To enhance the performance, the researchers combined feature extraction strategies with intelligent algorithms, transforming the problems with high dimensions into the low-dimension space. At present, the tentative application of intelligent algorithms based on feature extraction yielded concrete achievements. The

principal component analysis based genetic algorithm (PCAGA) (Zhang et al., 2017a), the modified artificial bee colony algorithm (MABC) (Ren et al., 2016), the dynamic search fireworks algorithm with linearly decreased dimension number strategy (ld-dynFWA) (Mu et al., 2017a) and PCA-based flower pollination (PCAFP) (Yuan et al., 2016) have been successfully applied in the researches of tropical cyclone adaptive observations, El Niño-Southern Oscillation, and double-gyre variation, respectively. The CNOPs obtained by these methods have similar patterns and larger fitness values in comparison to the adjoint

method. It is indicated that the PCA-based intelligent algorithm is appropriate for solving CNOP in high dimensional numerical models, especially the models without the adjoint model, like CESM.

The objective of this paper is to find the OPRs which can produce the NAO anomaly pattern and explore the effect of the nonlinear process. We study the topic using the CESM, which is an ocean-atmosphere coupled model without an adjoint model. Thus, traditional algorithms like SPG2 are inappropriate for this case. The particle swarm optimization (PSO) and

genetic algorithm (GA) hybrid algorithm (PSO-GA) (Kumar and Vidyarthi, 2016; Agarwal and Srivastava, 2018), which is combined with the principal component analysis (PCA) strategy (PGAPSO), is selected to identify OPRs of the NAO. NAOI is selected as the objective function (North Atlantic sector), while the perturbations are superimposed on the Arctic region. We adopt total dry energy over the Arctic atmosphere as the constraint condition. The variation of NAOI and SLP are traced during simulation time, and the OPRs derived from the PGAPSO are compared with other adjoint-free methods (BGM, random) in

terms of pattern structure and fitness values for validation. It is found that the nonlinear process mainly plays a role in the last days of the optimization time. For two cases (Case 1 and Case 2 in Section 4) in different initial conditions, OPRs obtained by the improved algorithm can steadily turn the basic state into anomaly mode with anomaly higher (lower) NAOI. The structures of OPRs are also explored. It shows that the wind over the Arctic Ocean as well as the temperature around Greenland have obvious characteristics and a noticeable impact for $NAO^+$ and $NAO^-$. The temperature is discovered as an important factor





to trigger NAO events. In addition, the solving system is accelerated with multiple parallel frameworks. After parallelized with MPI and CUDA, the speed-up ratio of the intelligent solution system reaches $40.2\times$ compared with its serial version.

   The structure of this paper is organized as follows: Section 2 describes the CESM, and Section 3 presents the CNOP method, the PGAPSO algorithm, and the parallelization technique. Experiments and results are displayed in Section 4. This paper ends
with a conclusion and future work in Section 5.

## 2   CESM

The CESM (Kay et al., 2015) is a new generation of fully coupled climate model developed in 2010. It has been widely used to simulate the carbon cycle (Lehner et al., 2015), ocean currents (Large and Caron, 2015), soil moisture (Swenson and Lawrence, 2012), precipitation (Hagos et al., 2016) and other climate phenomena. As shown in Figure 1, the CESM is composed of seven
geophysical model components, respectively Atmospheric (Community Atmosphere Model, CAM), Sea-ice (CICE), Land (Community Land Model, CLM), River-runoff (River Transport Model, RTM), Ocean (Parallel Ocean Program, POP), Land-ice (CISM), Ocean-wave (XWAV). The CESM also has a coupler (CPL) that coordinates the time evolution of geophysical models and delivers information between these components.

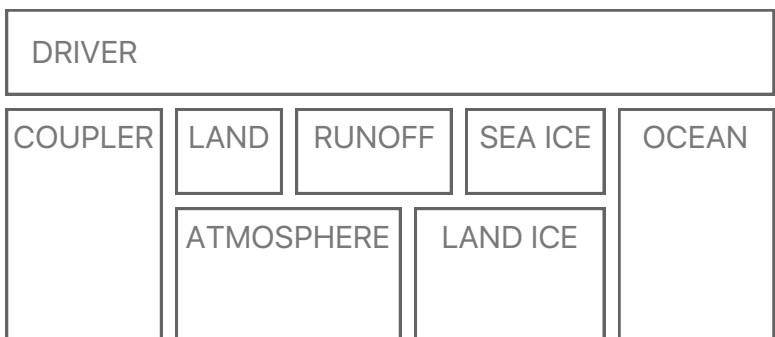

**Figure 1.** Main components of CESM.

   The atmospherical component in CESM 1.2.2 is used to simulate the NAO in this work. CAM version 5.3, which is a
global atmospherical general circulation model developed from the NCAR CCM3, is released as the atmosphere component of CESM 1.2. The CAM incorporates an interactive aerosol model where aerosols interact with the tropospheric chemistry. The component set we selected is *F_2000* that includes CAM, CLM, and CICE (prescribed mode) activated with SST data mode, and the CAM component is stand-alone.

   We simulate the NAO cases in the winter (DJF), and the parameter *nhtfrq* is set to -24, which denotes the daily average. The
experiments are performed on a resolution of *f09_g16* with an approximate grid spacing of $0.9°\times1.25°$, including 26 levels in the vertical. Perturbations are superimposed on the basic state of Arctic region (between $60°N$ and $90°N$, see Figure 2) and contains six variables listed in Table 1. The size of $U$, $V$, $T$, and $Q$ is 26 (layer) $\times$ 32 (latitude) $\times$ 288 (longitude), and the size




of $PS$ and $PHIS$ is $32 \times 288$. The NAOI increment is selected as the fitness value and is calculated by the output variable SLP. The region of SLP mainly locates between $20°N$ and $80°N$ and between $90°W$ and $40°E$, as shown in Figure 2. The vector of SLP only has one layer and consists of $65 \times 105$ grids.

**Table 1.** The related variables included in the perturbations.

| Variable name | Description | Units |
|---|---|---|
| $U$ | Zonal wind | $m/s$ |
| $V$ | Meridional wind | $m/s$ |
| $T$ | Temperature | $K$ |
| $Q$ | Specific humidity | $kg/kg$ |
| $PS$ | Surface pressure | $Pa$ |
| $PHIS$ | Surface geopotential | $m^2/s^2$ |

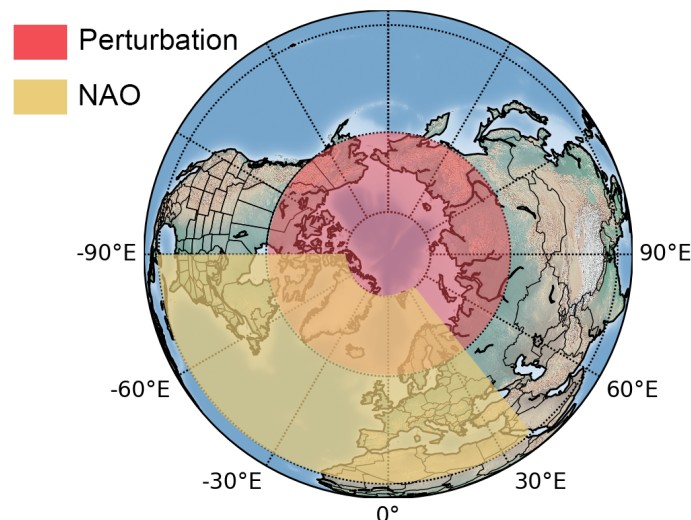

**Figure 2.** The regions of perturbations and the NAO. The perturbations superimpose on the Arctic section between $60°N$ and $90°N$ (shown in red). The region of NAO is between $20°N$ and $80°N$ and between $90°W$ and $40°E$ (shown in yellow), and it is also the scope of the SLP variable.





## 3 CNOP and PGAPSO

### 3.1 CNOP

The CNOP is a natural extension of the linear singular vector into the nonlinear regime and is proposed to study predictability problems of weather and climate in numerical models (Mu et al., 2009). The CNOP can represent the initial perturbation that

can trigger the largest uncertainty at the prediction time. The OPR of NAO is a kind of CNOP that makes the basic state evolve into the NAO events. Suppose the nonlinear model can be briefly described as follows:

$$\frac{\partial S}{\partial t} + F(S) = 0, S|_{t=t_0} = S_0 \tag{1}$$

where $S$ denotes the state vector, and $S_0$ is the basic state at the initial time $t_0$. F is the nonlinear partial differential operator of the model. The equation (1) has the discrete form:

$$S_t = M_{t_0 \to t}(S_0) \tag{2}$$

where $M_{t_0 \to t}$ represents the nonlinear propagator that "propagates" the initial state from time $t_0$ to the prediction time $t$. $S_t$ is the state vector at time $t$. If we superimpose an initial perturbation $s_0$ on the basic state, the development of the initial perturbation would be:

$$\Delta S = S_t^{'} - S_t = M_{t_0 \to t}(S_0 + s_0) - M_{t_0 \to t}(S_0) \tag{3}$$

The OPR refers to the initial perturbation that can make the objective function achieve the maximum (minimum) under the constraint condition at prediction time. In this work, the NAOI difference between perturbation state and the reference state is selected as the objective function $J$:

$$J(s_0^*)_{NAO^+} = \max_{\|s_0\| \le \epsilon} J(s_0) = NAOI(NAO^+)_{CNOP} - NAOI_{refer} \tag{4}$$

$$J(s_0^*)_{NAO^-} = \min_{\|s_0\| \le \epsilon} J(s_0) = NAOI(NAO^-)_{CNOP} - NAOI_{refer} \tag{5}$$

where $s_0$ is the initial perturbation and consists of physical variables listed in Table 1. According to the definition of the OPR, the perturbation $s_0^*(NAO^+)$ is the OPR of the $NAO^+$ and makes $J$ achieve the maximum, whereas $s_0^*(NAO^-)$ is the OPR of the $NAO^-$ and makes $J$ achieve the minimum. $\epsilon$ denotes the constraint condition of the OPRs, ensuring the perturbation within a reasonable range. The constraint condition is consulted from the related works of the sensitive area identification for





tropical cyclone (Zhang et al., 2017a) owing to the same variables, and is set to 10% of the summation of the kinetic energy of basic state in the vertical coordinate $\sigma$ and verification areas $D$:

$$\epsilon = 10\% * \frac{1}{D} \int_D \int_0^1 [U_0{}^2 + V_0{}^2 + \frac{C_p}{T_r}T_0{}^2 + R_a T_r (\frac{\Pi_0}{\pi_r})^2] d\sigma dD \tag{6}$$

where $U_0$, $V_0$, $T_0$, and $\Pi_0$ are the initial conditions of zonal wind, meridional wind, temperature, and surface geopotential

respectively. $C_p$ is the specific heat at the constant pressure which is set to 1005.7 $J \cdot kg^{-1} K^{-1}$ and $T_r$ is the reference temperature with a value of $270K$. $R_a$ denotes the ideal gas constant, and its value is set to 287.05 $J \cdot kg^{-1} K^{-1}$. $\pi_r$ is the static reference pressure with a value of 1000 $hPa$.

In this experiment, we choose a blocking indicator proposed by Liu (Liu, 1994) to quantify the extent of the NAO events. The NAOI is defined as the projection of the SLP field on the NAO anomaly pattern:

$$NAOI = \frac{\langle SLP_{NAO}, SLP_d \rangle}{\langle SLP_{NAO}, SLP_{NAO} \rangle} \tag{7}$$

where $SLP_d$ is obtained by subtracting the climatological mean from SLP output, and $\langle \rangle$ denotes the inner product operation of vectors. $SLP_{NAO}$ denotes the NAO anomaly pattern decomposed by the empirical orthogonal function (EOF) analysis, which is illustrated in Figure 3. The NAO spatial pattern is manifested as a typical meridional dipole mode, including the Iceland low pressure along with the North Atlantic subtropical high. In Figure 3, it is a pattern of the $NAO^+$, presenting the mode with the

negative anomalies in high latitude and the positive anomalies in low latitude.

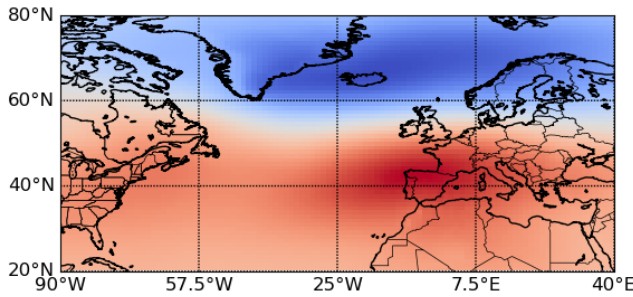

**Figure 3.** The first mode of the EOF with SLP anomaly field concentrated in the North Atlantic region between $90°W$ - $40°E$, $20°N$ - $80°N$.





Combining the equation (4), (5) and (7), the objective function is described as follows:

$$J(s_0) = \Delta NAOI$$
$$= \frac{\langle M_{t_0 \to t}(S_0 + s_0) - M_{t_0 \to t}(S_0), SLP_{NAO} \rangle}{\langle SLP_{NAO}, SLP_{NAO} \rangle} \tag{8}$$

In summary, the objective function is the projection of the SLP field difference between the final state and the reference state on the NAO anomaly pattern, and the OPR of $NAO^+$ ($NAO^-$) can be obtained by solving the maximum (minimum) of the

objective function $J$.

## 3.2  PGAPSO

Under the resolution of $0.9° \times 1.25°$, the total dimensions of variables involved in the objective function are 4 ($U, V, T, Q$) $\times$ 26 (layer) $\times$ 32 (latitude) $\times$ 288 (longitude) + 2 ($PS, PHIS$) $\times$ 32 (latitude) $\times$ 288 (longitude) = 976896. It is difficult for the algorithm to search for the optimal solution in such high dimensions. Thus, we reduce the scale of the solution domain via

eigenvector decomposition using the PCA method.

First, we obtain 900 original samples after running 10-year integration (only in winter) on a daily average using CESM. Owing to their different physical units, we need to perform Z-Score normalization for each variable:

$$P_v = \frac{P_v - \overline{P_v}}{\sigma_v} \ (v = 1, 2, \ldots, c) \tag{9}$$

where $c$ refers to the number of variables. $\overline{P_v}$ is the mean value of the variable $v$, and the $\sigma_v$ denotes the standard deviation.

After that, each piece of sample is reshaped into a vector with one dimension (1 $\times$ 976896), these samples constitute a matrix with a size of 900 $\times$ 976896. Then each sample is subtracted from the mean values of these samples for centering, and the obtained sample is weighted according to the area of the grid:

$$P_{ij} = (P_{ij} - \frac{1}{n} \sum_{i=1}^{n} P_{ij}) * \cos(lat(j)) \ (i = 1, 2, \ldots, n; j = 0, 1, \ldots, r) \tag{10}$$

where $P_{ij}$ denotes the $j^{th}$ element of the $i^{th}$ sample data. $n$ is the number of the samples, and $r$ is the length for each sample.

$lat(j)$ is the corresponding latitude of the $j^{th}$ element, and the area weight is calculated approximately via the cosine value of the latitude. Then the eigenvalues ($\lambda_1, \ldots, \lambda_n$) and eigenvectors of the covariance matrix $PP^T$ are calculated to obtain principal components:

$$PP^T L = L\Sigma \tag{11}$$

where $L$ is the eigenvector matrix, and $\Sigma$ is a diagonal matrix whose entries in the main diagonal are the corresponding

eigenvalues. The top $m$ columns of the eigenvectors $L$ sorting by their eigenvalues are selected as the principal components. The





value of $m$ is determined by the contribution rate. $m$ is set to the minimum number of dimensions that meets the contribution rate of 90% to balance the efficiency and effectiveness. The reduced space with $m$ dimensions is far smaller than the original one.

To obtain the extremum of the objective function, we adopt a hybrid algorithm improved from two efficient algorithms, PSO and GA. The PSO is a type of intelligent heuristic algorithm to solve the problem with NP property (Kennedy, 2011). The position with the best fitness value is searched by tracing both individual optimal position and the optimal global position. The algorithm flow can be briefly described as follows: (1) Initialize the speed ($V$) and position ($X$) of each particle with random values. The random values obey the normal distribution and ensure most of the perturbations satisfy the constraints. (2) For each particle $i$, position vector is in reduced space, so the position vector needs to be restored into original space via $X_i' = X_i \cdot L_{1,...,m}$, thereinto, $L$ is an $m \times m$ eigenvector matrix. Then superpose the perturbation $X_i'$ on the basic state. When the model integration is finished, calculate the fitness value of each particle through the equation (8) and record its optimal position ($X_{pb}$) along with the optimal global position ($X_{gb}$). (3) Update the position and speed of each particle. (4) Repeat steps as mentioned above until satisfying the terminal condition. The updating formula is as follows:

$$
\begin{cases}
V_i^{k+1} = \omega_k V_i^k + c_1 r_1 (X_{pb}^k - X_i^k) + c_2 r_2 (X_{gb}^k - X_i^k) \\
X_i^{k+1} = X_i^k + V_i^{k+1}
\end{cases}
\tag{12}
$$

where $V_i^k$ is the speed of particle $i$ for step $k$ and $V_i^{k+1}$ is for particle $i$ at step $k+1$. $c_1$ is the self-awareness coefficient for the historical self-optimal position, and $c_2$ is the social-awareness coefficient for the optimal global position of all particles. The empirical value of $c_1$ and $c_2$ are both set to 2. $r_1$ and $r_2$ are random factors in range of $[0,1]$. $X_{pb}^k$ refers to the best position of particle $i$ up to the step $k$, and $X_{gb}^k$ represents the best position of the entire swarm in $k$ steps. $\omega_k$ is the weight parameter and is calculated by:

$$
\omega_k = \omega_{max} - \frac{\omega_{max} - \omega_{min}}{iter_{max}} * iter
\tag{13}
$$

where $iter$ is the current number of steps, and $iter_{max}$ is set to 50.

PSO is the main body of the PGAPSO, and the GA further optimizes the search process. As a metaheuristic algorithm, the GA derives from natural selection (Goldberg and Holland, 1988). When the speed norm is smaller than the threshold, the particles are fed into GA for further search. The main operations of GA include selection, crossover, and mutation. Particles are selected according to their fitness value to breed a new generation. The selection probability for each individual is equal to the ratio of its fitness value to the sum of the fitness value population:

$$
p_s = \frac{J(s_{X_i'})}{\Sigma J(s_{X'})}
\tag{14}
$$





After that, the selected parents generate new individuals via crossover:

$$X_a^{'}\{x_s,\ldots,x_e\} = X_b\{x_s,\ldots,x_e\}$$
$$X_b^{'}\{x_s,\ldots,x_e\} = X_a\{x_s,\ldots,x_e\}$$

(15)

Then the new generation mutates in a single location with probability $p_m$ to increase the randomness. The fitness value of each new particle is compared against its parents, and the best position is recorded. If the new individual has a better fitness value than the global best position, the global best position ($X_{gb}^k$) would be replaced. Then the speed and position of particles are updated using equation (12), and the final global fitness value is obtained until the $iter$ reaches $iter_{max}$.

## 3.3 Parallelization

The computation of CNOP in CESM is quite time-consuming. With 48 CPU cores, 30 particles, and 50 iterations, it takes about 8.33 days to obtain the OPRs in the serial program. Multiple parallel techniques and frameworks are adopted to improve the performance in this work.

### 3.3.1 CESM Parallelization

The role of the CAM component in CESM is to simulate the variation of atmosphere, and the integration process involves plenty of matrix operations with high-dimensional input data. Thus, the model simulation becomes the bottleneck of the program performance. Although CESM has already been parallelized using Message Passing Interface (MPI) and Open Multi-Processing (OpenMP), it is still time-consuming.

Recently, the Graphics Processing Unit (GPU) has been widely used in accelerating numerical models. Since GPU is suitable for large-scale parallel computing, it can significantly improve the execution performance of numerical calculation in climate models. A parallel scheme for Community Climate System Model (CCSM) has been proposed to shorten the runtime of climate prediction by porting the radiation module onto GPUs (Coleman and Feldman, 2013). The module was optimized using the inline method and MPI routines. A cloud analysis scheme called Goddard Cumulus Ensemble (GCE) in Weather Research and Forecasting (WRF) was highly optimized using NVIDIA Tesla K40 with 2880 cores (Huang et al., 2015). Compared to the CPU-based parallel version running on four nodes, the GPU-based scheme performed better. A novel asynchronous strategy has provided significant performance benefits in CESM (Korwar et al., 2013). The most time-consuming routines have been accelerated via OpenACC directives, and it achieves a speedup of 1.19×-1.53× for the entire model. Another attempt for accelerating CESM was to port CESM along with a rewritten vertical remapping scheme onto GPUs (Carpenter et al., 2013).

Previous work results have shown that the performance of the optimized subroutine can be improved by the GPU technique substantially. In this work, we port several time-consuming subroutines in CAM onto GPUs via the PGI CUDA Fortran platform. The subroutine runtime is analyzed using *pref*, which is a performance evaluation tool for Linux. As shown in Figure 4, subroutine *radclwmx* and *radabs* both execute longer compared with other subroutines, so we rewrite these two subroutines with the CUDA interface. Simultaneously, we adopt kernel directives and OpenACC directives to simplify specific operations





on the device. Multiple asynchronous streams are created to overlap function execution and data replication. Moreover, the loops in subroutines are reconstructed to minimize I/O transfer times. For the compilation phase, the option $-O4$ is selected to perform the optimization at the highest level. The option *-fast* and *-fastsse* are also utilized to launch the 64-bit Single Instruction Multiple Data (SIMD) instruction and implement cache alignment and flush.

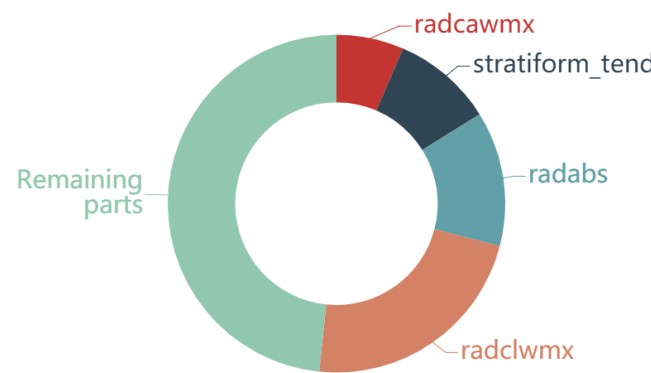

**Figure 4.** Execution time division of CAM subroutines.

### 3.3.2 PGAPSO Parallelization

In each iteration of PGAPSO, the calculation of fitness value for each particle is relatively independent. According to its features, multi-process techniques are suitable for executing these tasks concurrently. Here we adopt MPI, which enables the parallelization of the program by supporting communication and broadcasting between nodes, as the parallel framework to accelerate the algorithm. The pseudocode describes the flow of the parallel PGAPSO:

Assume that $n$ particles are initialized, then $n+1$ processes are launched. These processes are divided into two groups: one is a master process, the others are slave processes. We assign a process for each particle as the slave process so that CESM can be called in parallel, and the fitness value can be obtained simultaneously. Figure 5 illustrates the parallel architecture of PGAPSO for solving CNOP. At each iteration, the master process allocates calculation tasks to slave processes. For each slave process, perturbation under constraint condition is generated and is superimposed on the basic state of CESM. Then CESM, which is

paralleled with MPI, OpenMP, and CUDA, is called to perform the integration. The fitness values are calculated via equation (8) and are gathered by the master process. After that, the master process broadcasts the optimal global value to slave processes via MPI. If particles' speed norm is less than the threshold value, the crossover and mutation operations are conducted. Then the master process compares the current particles and updates the information of optimal particle at the end of each iteration. With the help of MPI, the performance of PGAPSO can be significantly enhanced.





---

**Algorithm 1** Pseudo code for PGAPSO

---

1: transform training data through PCA to obtain principal components $L_d$ with $m$ dimensions

2: initialize population

3: **for** $iter = 1$ to $iter_{max}$ **do**

4:     **for** each process **do**

5:         restore solution matrix $X_i$ into original space via $X_i * L_d$

6:         calculate fitness value $J(s_0) = F(\|M_{t_0 \to t}(S_0 + P * L_d)\|)$ under the constraint $\epsilon$

7:     **end for**

8:     gather results from each process

9:     **if** norm of particle speed $\leq \xi$ **then**

10:         select individuals according to $\frac{J(s_{x_i'})}{\Sigma J(s_{x'})}$

11:         crossover and mutate with probability $p_m$

12:         compare fitness values between new generation and parent individuals

13:     **end if**

14:     update particle speed via $V_i^{k+1} = \omega_i V_i^k + c_1 r_1 (X_{pb}^k - X_i^k) + c_2 r_2 (X_{gb}^k - X_i^k)$

15:     update particle position via $X_i^{k+1} = X_i^k + V_i^{k+1}$

16: **end for**

---

## 4 Experiments and Results

### 4.1 Experimental Environment

We conduct experiments on the Tianhe-2 supercomputer, which is located in the National Supercomputer Center in Guangzhou, China. Each node consists of 2 Intel Ivy Bridge Xeon processors connected by Intel QuickPath Interconnect. NVIDIA Tesla

5  K80 GPUs on Tianhe-2 are used in our GPU-based scheme for CESM acceleration. Each Tesla K80 GPU has 4992 CUDA cores, and its double-precision performance is up to 2.91TFLOPS. Data transmission between CPUs and GPUs depends on PCI-e 3.0 bus with 40 lanes.

### 4.2 Dimensions of solution space

The first step is to decompose the principal component from the original sample. From the equation (11), we know that the

10  dimension of solution space equals to the number of eigenvectors selected according to the accumulative variance ratio. Fig. 6 displays the accumulative variance ratio of different component dimensions. It can be seen that the accumulative variance ratio increases as the dimension of principal components increases. To balance effect and performance, we select the top 50 eigenvectors as the principal components corresponding to the cumulative explained variance ratio of just over 90%. In other words, the OPRs are solved in a degenerate space with 50 dimensions.

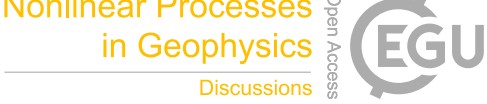

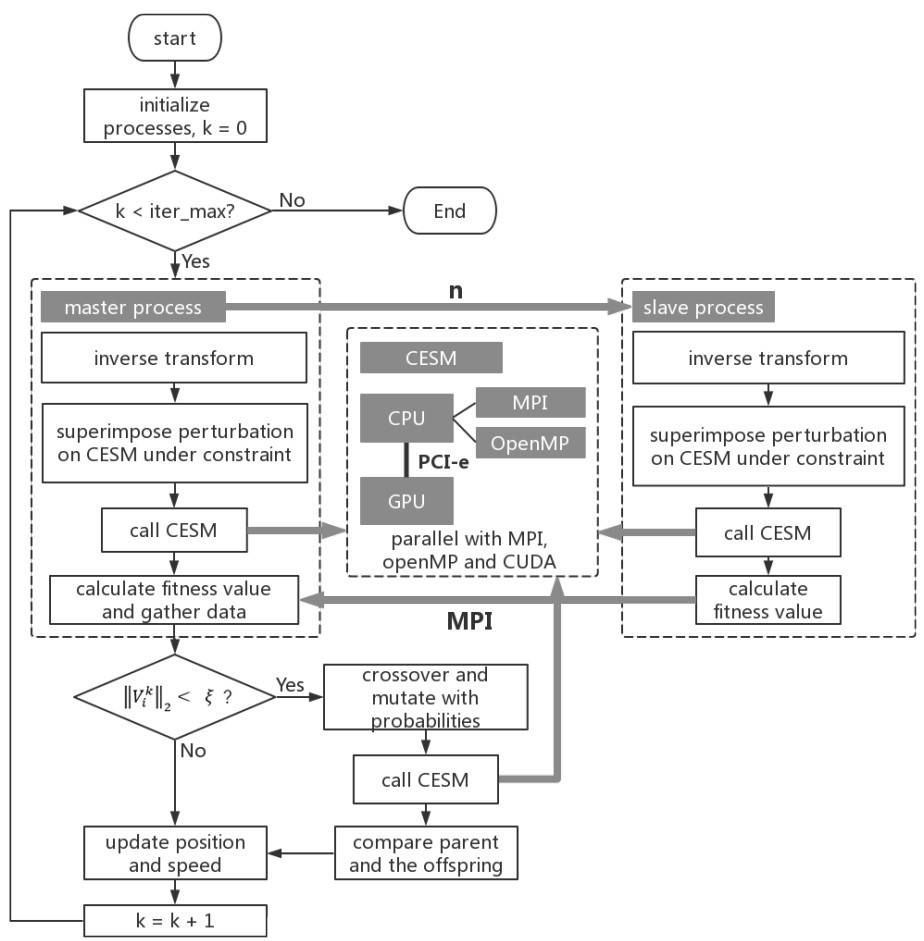

**Figure 5.** The parallel architecture of PGAPSO for solving CNOP.

## 4.3 Simulation Duration for OPR

Since the NAO has an approximated e-folding period of two weeks, we intend to determine the most appropriate simulation duration within 15 days. We select 5 days, 7 days, 10 days, and 15 days as the optimization times to observe the largest variation of the NAOI amplitude. Using the PGAPSO with 50 iterations, the distribution of the optimal fitness value at different

5    simulation durations is illustrated in Fig. 7. From the box plot, we can see that the 5-day simulation has the narrowest range of the $\Delta NAOI$, especially for $NAO^-$. The 7-day simulation has a wider variation range of values for $NAO^-$, and the 10-day optimization is more inclined to evolve into the $NAO^-$. By contrast, only the 15-day simulation can steadily trigger the NAO


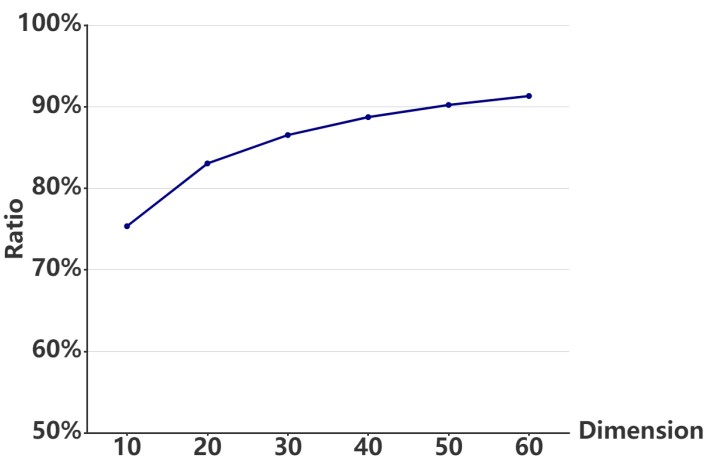

**Figure 6.** The accumulative variance ratio at different dimensions of principal components.

events for both two types of phases. The maximum and the minimum values of $\Delta NAOI$ are 2.23 and -3.91, which is much larger than 1 or less than -1.

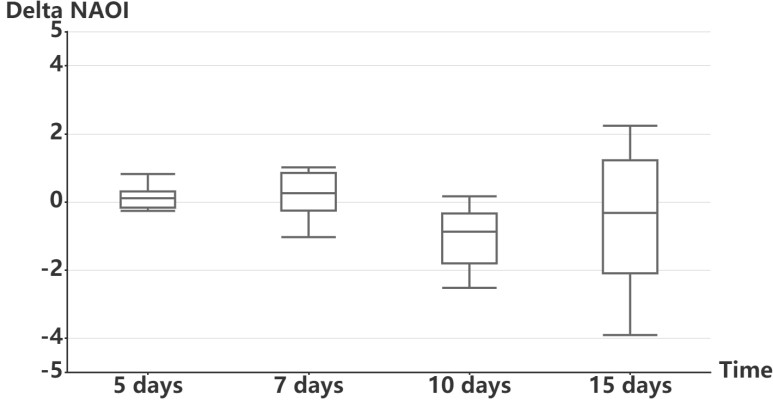

**Figure 7.** The distribution of optimal fitness values at different simulation durations.

Then we need to verify whether the OPRs obtained by PGAPSO can trigger the NAO events. We acquire the SLP difference patterns by subtracting the SLP data of reference state from the output, and these patterns for different simulation durations are
5   shown in Fig. 8. The patterns of 5-day and 7-day simulation are in a state of instability with multiple dispersed pressure centers. We can find that a strong negative center appears in the mode of $NAO^-$ in 7-day simulation, and this is the reason for the relatively lower $\Delta NAOI^-$ in Fig. 7. In the same way, the strong positive center in the pattern of 10-day simulation also causes the abnormal $\Delta NAOI^-$. The typical feature of NAO event is the dipole SLP mode located near Iceland and Azores. Although





the approximated structure of NAO events appears in the 7-day and 10-day optimization, the pressure cores are irregular and discrete with the trend of migration. As for the 15-day integration, the dipole centers form and migrate across the Atlantic Ocean, which is particularly evident in $NAO^-$. Therefore, we select 15 days as the simulation duration in our experiments.

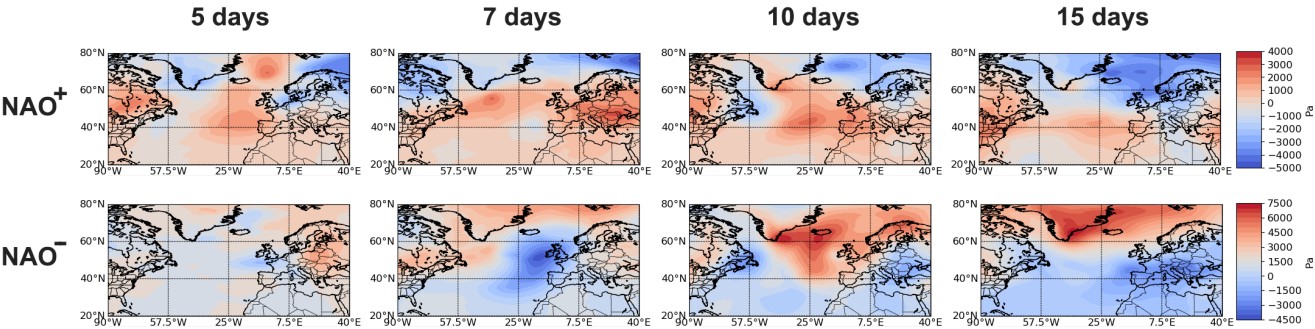

**Figure 8.** The difference of SLP patterns between perturbation state and reference state at different simulation durations.

### 4.4 Evolution of NAOI and SLP pattern

To visualize the effects of PGAPSO, we compare it with two adjoint-free methods in the cases of 15-day simulation. One of them is the breeding of growing mode (BGM), which has been put into operation by the National Centers for Environmental Prediction (NCEP) (Toth and Kalnay, 1997). The breeding vector (BV) generated from BGM retains the dynamical structure of rapid growth and is proven compatible with the atmospheric model. The crucial point is that it can be solved without gradient information. The other procedure is the random method, which has the same search times with the PGAPSO. The

NAOI amplitude flows for these methods are displayed in Fig. 9. $CNOP_{PO}$ and $CNOP_{NE}$ denote the two types of OPRs ($NAO^+$ and $NAO^-$) solved by the PGAPSO, which is based on the CNOP approach. In Fig. 9, the black dashed line refers to the reference state, which is the final state acquired without perturbations. We superpose perturbations on two different initial conditions (Case 1 and Case 2), and the CNOP-type perturbations perform better on both $NAO^+$ and $NAO^-$. Notably, only $CNOP_{PO}$ triggers the extremely high NAOI in Case 2. The BVs ($Bred_{PO}$ and $Bred_{NE}$) are also effectual for forming the structures that reflect the NAO events, particularly in $NAO^-$. Besides, the NAOI curves corresponding to $Bred_{PO}$ and

$Bred_{NE}$ have the similar varying tendency in the first 10 days and evolve into the contrary phases in the final days of the simulation duration. However, the $Bred_{PO}$ does not work well in whether Case 1 or Case 2. Since the search directions of random vectors are entirely random, they do not cause the apparent variation. The development of BVs and random vectors depend on the basic state to some extent, while CNOPs steadily make $|NAOI|$ are greater than 2 in both $NAO^+$ and $NAO^-$

and are almost not affected by the initial state.

Fig. 10 shows the SLP patterns using these methods in Case 1. For the $NAO^+$, although the center intensity is slightly weak in the pattern of PGAPSO, it presents the principal characteristics of $NAO^+$ events. The opposite pressure fields generate in the center of the Atlantic sector and are symmetrical about the $50°N$. In BV's pattern, a strong positive core appears in the


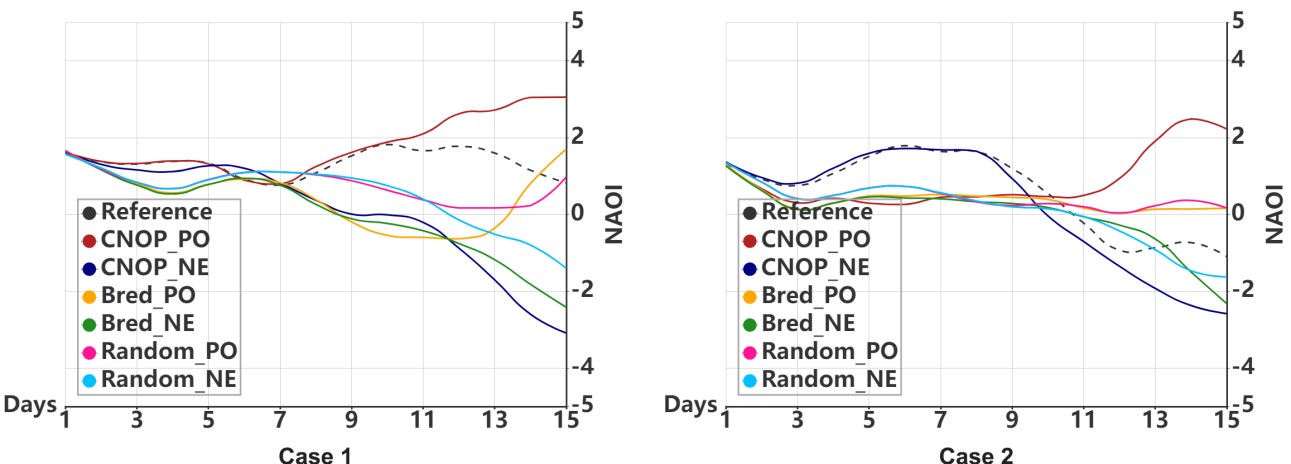

**Figure 9.** The trends of the NAOI amplitude for different methods and reference states.

center of the Atlantic Ocean, while the negative pressure field migrates to the Norwegian Sea, resulting in the overall structure tilts to the east. The pattern of random vector is almost occupied by positive pressure, which is mainly located in Irish islands. Unlike the $NAO^+$, the $NAO^-$ displays similar structures in these patterns, which have the typical NAO mode with a strong positive center located in Iceland and the negative center(s) around the Azores. The difference is that, the patterns of BV and

random vector include several discrete negative cores. In Case 1, the OPRs solved by PGAPSO can trigger two types of NAO events with typical dipole mode.

The situation in Case 2 is illustrated in Fig. 11. For $NAO^+$, the OPR obtained by PGAPSO forms an SLP pattern with noticeable features of the $NAO^+$ event, while BV and random vector generate irregular SLP fields with scattered centers. From Fig. 9, we find that the perturbation state has a larger probability of moving to the contrary phase of the reference. while

CNOPs can still make the basic state evolve into the NAO event that has the same phase with the reference state.

To observe the evolution of the NAO events during the 15-day simulation period, we plot the SLP difference field on Day 1, Day 5, Day 10, and Day 15 in Fig. 12. It can be seen from the figure that the SLP difference patterns do not show any changes on the first day—just after the perturbations are superimposed. On the Day 5, the difference is still unnoticeable, and a relatively concentrated pressure center appears in the Greenland Sea only in the mode for $NAO^-$. As of the Day 10, the change

in pressure difference spreads outward from the center of the Arctic, and the area of the negative (positive) pressure field in the Greenland Sea widens, with an increase in intensity. At the prediction time (Day 15), as the $NAO^+$ event occurs, the positive pressure over the Arctic Ocean reveals a regime shift from Day 10, and the negative pressure field around Iceland evolves into a strong one. The positive pressure field on the North Pacific Ocean also has an increasing trend. For the $NAO^-$ event, the positive cores around the North Pacific Ocean gather toward the North Pole, and the positive pressure center on Iceland and

Greenland gradually expands and reaches its peak. In this process, the most obvious stage of change is from Day 10 to Day



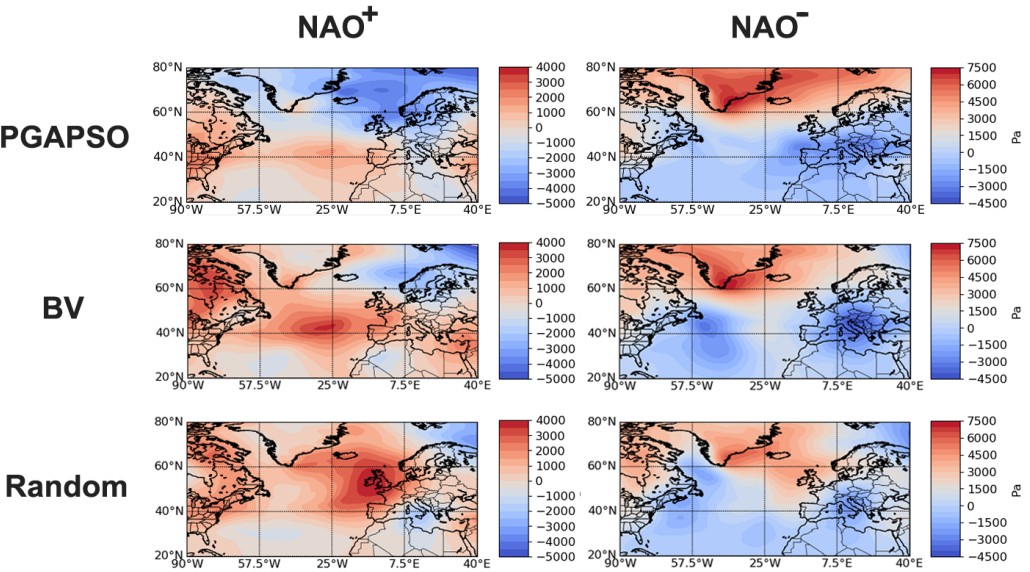

**Figure 10.** The SLP anomaly patterns obtained by multiple methods for $NAO^+$ and $NAO^-$ in Case 1.

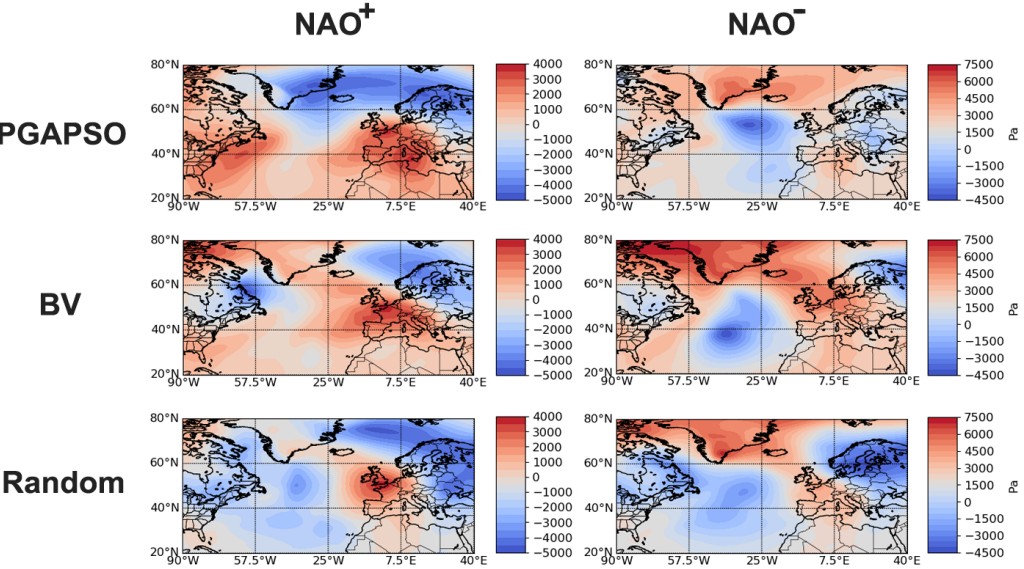

**Figure 11.** Similar to Figure 10, but for Case 2.

15, and the large difference of pressure and the dipole mode are formed at this stage. This also confirms the conclusion that the role of the nonlinear process is mainly at the end of the prediction phase.

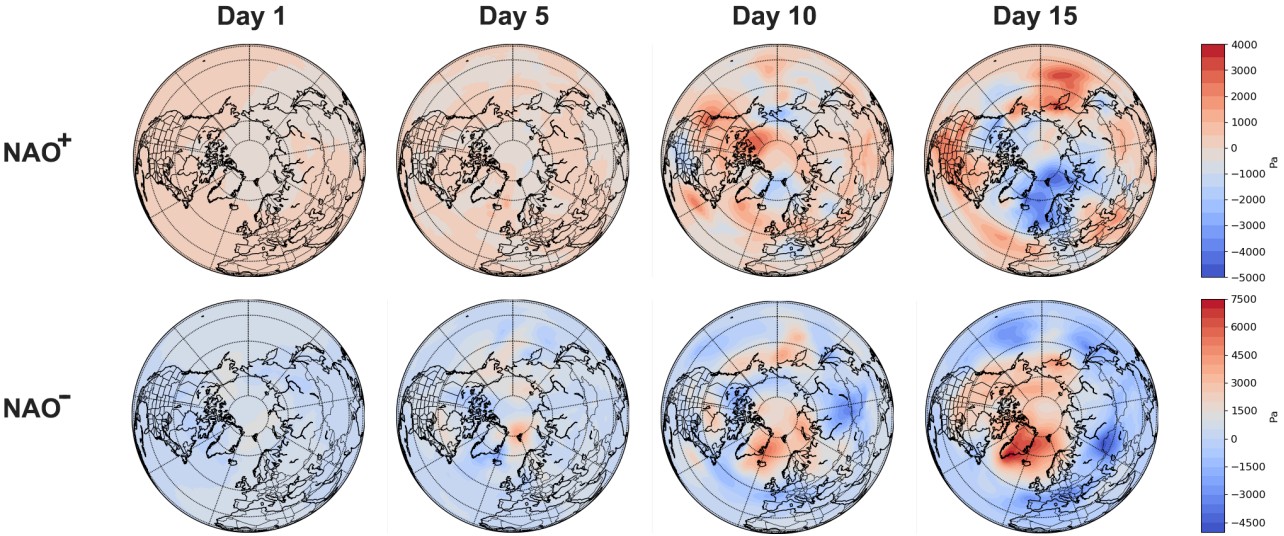

**Figure 12.** The nonlinear evolution on Day 1, Day 5, Day 10 and Day 15 at SLP field (Pa) with a simulation time of 15 days.

### 4.5 CNOP pattern

In this section, we aim to explore the structure of the OPRs that trigger the NAO event. Fig. 13 shows the OPRs obtained using the PGAPSO method for Case 1 in the previous section, including zonal wind, meridional wind, and temperature in the near-surface layer. The left subfigure is the OPR corresponding to the $NAO^+$, and the positive temperature structure appears in the Arctic Ocean region, accompanied by the wind from the Arctic to the Pacific Ocean. A negative temperature field distributes near Iceland, with the wind perturbation direction from the Arctic to the Atlantic. The wind direction of negative-phase OPR is opposite to that of positive-phase OPR in the Arctic Ocean and near Iceland, and the intensity of temperature is stronger than that of positive-phase OPR.

For Case 2, the OPR of the $NAO^+$ has a higher temperature intensity, while the OPR of the $NAO^-$ has smaller values in temperatures, along with the monotonous distribution characteristics. As can be seen from Fig. 9 and Fig. 10, under the circumstance of the relative high reference NAOI in Case 1, the amplitude of the OPR for $NAO^+$ is tiny compared to the $NAO^-$, and the initial state after perturbations superposition is more liable to develop into a $NAO^-$ state. The situation in Case 2 is similar to Case 1. The reference NAOI is in a negative phase, and the perturbation that can trigger a strong $NAO^+$ event has a larger intensity. This could mean that the simulation result is more inclined to migrate to the opposite phase as the reference state. It can also be seen from Fig. 13 and Fig. 14 that, for the two cases whose reference states are widely different, the patterns and value ranges of the OPRs are also substantially different.

The above diagrams describe the spatial characteristics of multi-variable perturbations, and the NAO events can also be triggered by a single variable, such as temperature. Following the above procedure, temperature perturbations are limited under a constrained condition of $T'^2 \leq 100$ and are superimposed on the $25^{th}$ layer of the atmosphere (near the surface) in the


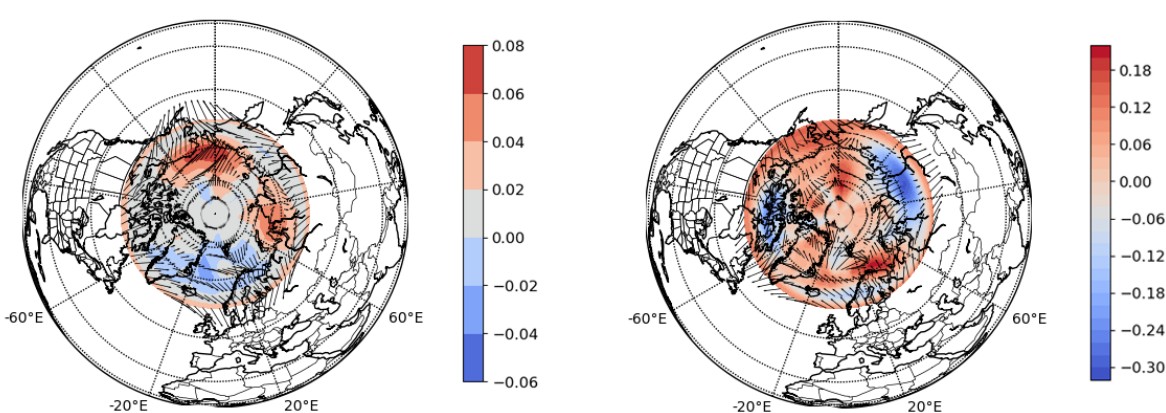

**Figure 13.** The OPRs for $NAO^+$ (left) and $NAO^-$ (right) in Case 1, consisting the temperatures and the winds.

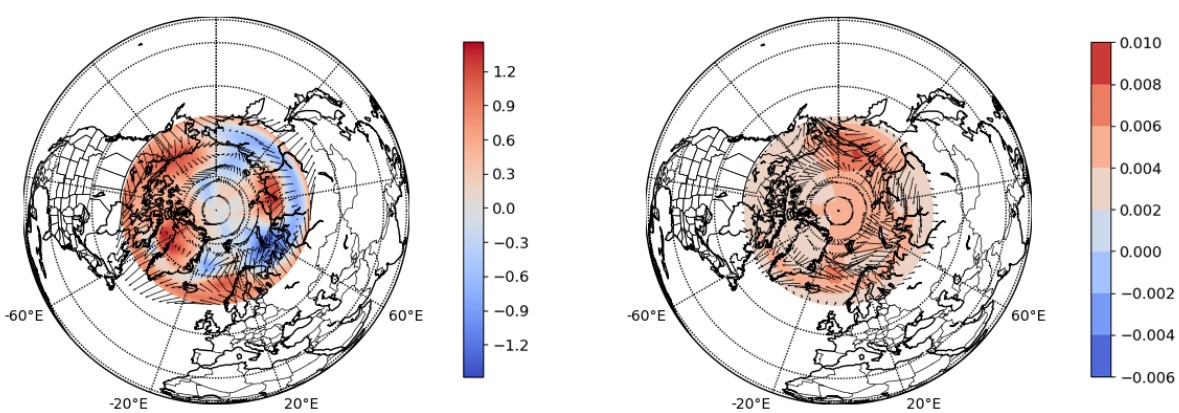

**Figure 14.** Similar to Figure 13, but for Case 2.





Arctic region. Using the PGAPSO, the OPRs composed of temperature are obtained, as illustrated in Fig. 15. The patterns of OPR of $NAO^+$ and OPR of $NAO^-$ have almost converse structures in the North Atlantic sector. There exists a noticeable pressure difference between Greenland and Iceland, with several centers in the mid-to-high latitudes and small cores around the Arctic region. Besides, the positive anomaly in eastern Europe is also conducive to the formation of the dipole. It aligns with

5 the hypotheses that atmospherical temperature gradients will result in the anomalous poleward atmospherical heat transport and an increased probability of the NAO reaching its high-index state.

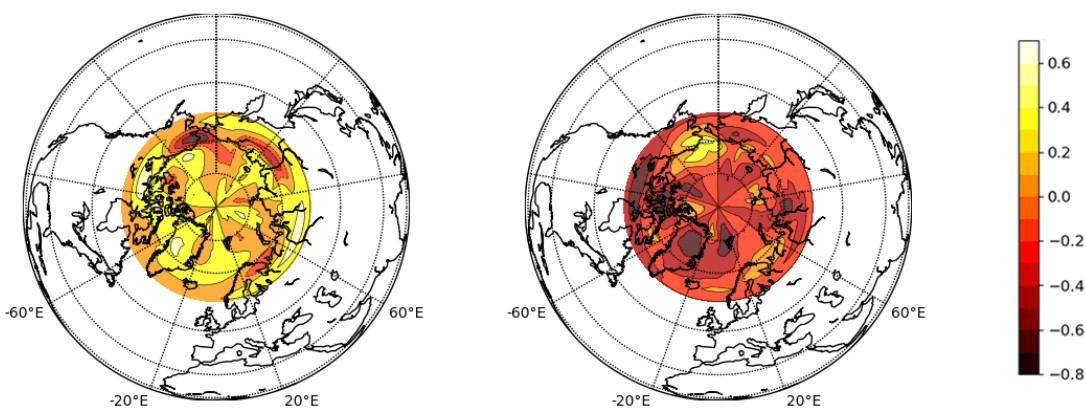

**Figure 15.** The temperature OPRs superimposing on the Arctic region for $NAO^+$ (left) and $NAO^-$ (right).

## 4.6 Performance Analysis

To manifest the performance improvement of parallel PGAPSO, Fig. 16 compares the runtime of parallel PGAPSO and serial PGAPSO in one iteration. Due to the limitation of computing resources, we conduct experiments on no more than 1080 CPU

10 cores. From Fig. 16, we can see that when the number of CPU cores is more than 720, it will take longer to run the serial algorithm. When the overmuch CPU cores are assigned to the serial program, the frequent communications would make the runtime increase. In contrast, the parallel version can make full use of resources, and its execution time keeps going down. The speedup ratio, which raises with the increasing CPU cores' number, is displayed in Table 2. With 1080 CPU cores, PGAPSO, based on the parallel scheme, achieves a speedup of $40.2\times$ compared to its serial version.

**Table 2.** The speedup of parallel PGAPSO compared with serial PGAPSO.

| Number of CPU cores | 240 | 480 | 720 | 840 | 960 | 1080 |
|---|---|---|---|---|---|---|
| Speedup ratio | $16.5\times$ | $17.9\times$ | $18.7\times$ | $26.3\times$ | $33.2\times$ | $40.2\times$ |


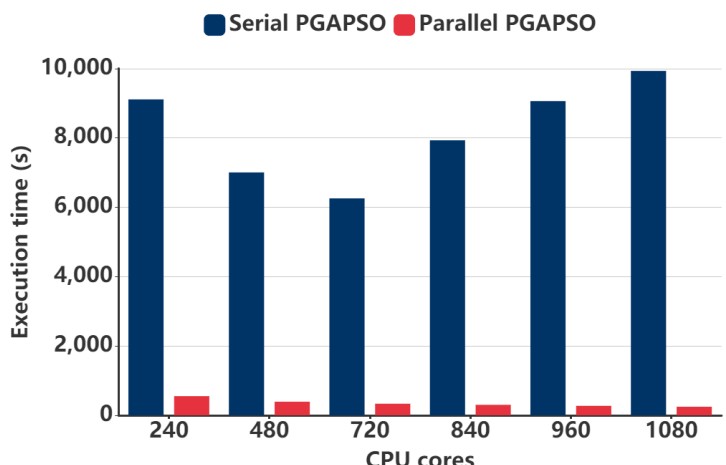

**Figure 16.** The execution time of serial PGAPSO and parallel PGAPSO with different number of CPU cores.

The parallel PGAPSO with GPU technique is also compared against the parallel PPSO and the parallel PGAPSO without GPU in Fig. 17. From Fig. 17, along with the increase of CPU cores, all methods have a trend of decrease in time consumption. When the CPU cores are increased to 1080, the runtimes of these three methods for one step are 334.12s, 270.18s, and 247.33s, respectively. Since the PGAPSO contains the operations like crossover and mutation, it may take slightly longer to execute

compared with the PPSO. However, the accelerators in GPU relieve the performance bottleneck caused by massive matrix computation in CESM. Although only two subroutines are optimized, the PGAPSO obtains promising speedup effects. It is also proved that the GPU has a potential capacity in accelerating numerical models.

Meanwhile, the convergence and optimal values of the PGAPSO are shown in Fig. 18. The speed norm is a measure indicator of the distance between a particle and the current optimal particle. Here we adopt mean speed norm to represent the convergence

of the entire swarm. If the mean speed norm approaches zero, it shows that all particles move to the positions close to the optimal particle, and it means the algorithm converges. In Fig. 18, there is not much difference between the PGAPSO and the PPSO at the beginning. The mean speed norm of the PGAPSO falls rapidly around Step 15 and converges in about Step 26. Besides, the best fitness value of the PGAPSO is greater than the PPSO. The advantages of the hybrid algorithm display in two aspects: the crossover operation of the GA has a relatively larger probability of generating particles with better fitness value,

and the mutation operation increases the randomness of the current particle to avoid plunging into local optimum. From Fig. 18, it is found that the PGAPSO improves the convergence speed and the solution quality compared with the PPSO.


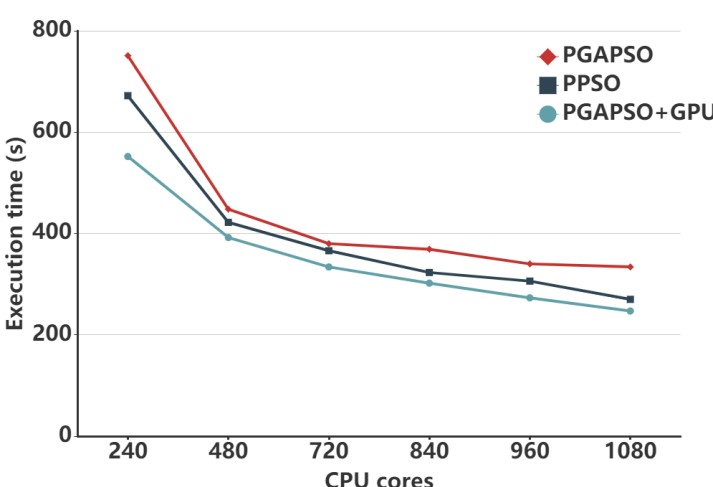

**Figure 17.** Time consumption of PGAPSO, PPSO and PGAPSO (GPU) with the growth of CPU cores.

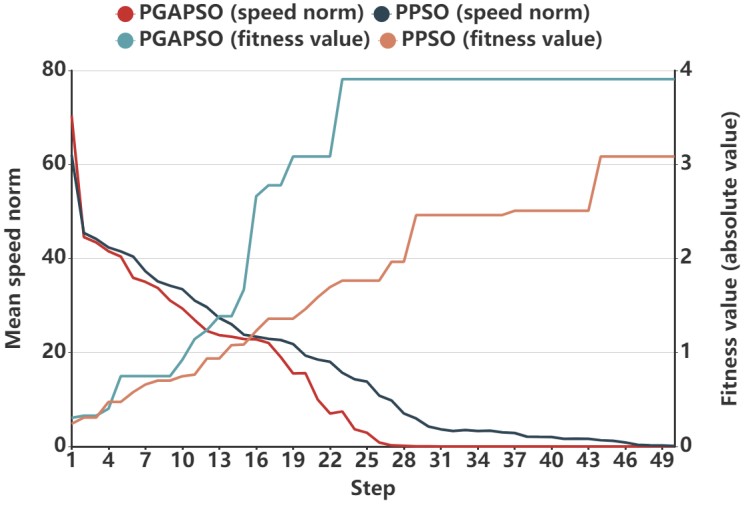

**Figure 18.** The mean speed norm of particles and global optimal fitness value of PGAPSO and PPSO in 50 steps.



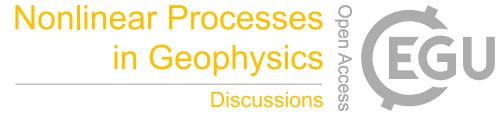

# 5   Conclusions

Initial condition errors are critical factors that result in uncertainty when simulating and predicting the NAO, and the NAO simulation can be improved by reducing the errors in initial condition. As a type of initial condition error, the OPR would cause the largest prediction error and eventually evolve into climate events. Therefore, the research of OPRs would help to
reveal the dynamic processes related to the NAO events and improve the prediction accuracy.

In this paper, we adopt a novel CNOP-based approach to study OPRs of the NAO using CESM. Since the CESM does not have a corresponding adjoint model, we cannot solve CNOP through the adjoint-based method mentioned in the previous works, such as SQP and SPG2. We present a hybrid intelligence algorithm named PGAPSO to solve CNOP. First, the optimization time are determined from experiments. Then the perturbations, which contain the variables of zonal wind, meridional
wind, temperature, specific humidity, surface pressure, and surface geopotential, are superimposed on the basic state. We investigate the OPRs in two cases with different initial state and similar NAOI, and both of the cases occur in winter (DJF). To validate the effectiveness of the PGAPSO, the trends of the NAOI amplitude and SLP anomaly patterns are compared to the BGM and random method. It is indicated that the OPRs obtained by PGAPSO can trigger the NAO events with the typical dipole pattern and have the largest $|\Delta NAOI|$. The SLP variation in the north hemisphere is traced during simulation time, and
the features of teleconnection patterns are identified via the SLP difference mode.

We also analyze the OPRs' structures for both $NAO^+$ and $NAO^-$ in these two cases. The wind directions of $NAO^+$ and $NAO^-$ present the opposite mode around Arctic Ocean and Iceland, and the temperature perturbations over Greenland island would promote the occurrence of NAO events. It is also demonstrated that even slight errors (see $NAO^-$ in Case 2) may cause a large uncertainty in simulation.

Since temperature has a significant effect on the NAO variation, the experiments with OPRs containing only temperature are conducted. Under a reasonable constraint, the temperature OPRs can still successfully trigger the NAO events for both $NAO^+$ and $NAO^-$. The OPRs for these two types of NAO events have almost converse structures, especially over Greenland and European, confirming the view of previous researches.

Moreover, multiple parallel frameworks are applied in this work to improve efficiency. The parallelization mainly consists
of two parts: parallelization of the algorithm with MPI and acceleration of CESM using CUDA. It significantly enhances the performance and achieves a speed-up of $40.2\times$.

Our future work is to apply the PGAPSO algorithm to study other climatological phenomena with the CNOP method. In this work, the CESM is regarded as a black box program. It is convenient to transplant the solver framework to other numerical models. We will also apply our approach to models that have high dimensions or have no corresponding adjoint model.

# Appendix A:  Abbreviations

**BGM**, breeding of growing mode
**BV**, breeding vector
**CAM**, Community Atmosphere Model



**CCSM**, Community Climate System Model

**CESM**, Community Earth System Model

**CICE**, Sea-ice

**CISM**, Land-ice

**CLM**, Community Land Model

**CNOP**, conditional nonlinear optimal perturbation

**CPL**, coupler

**CUDA**, Compute Unified Device Architecture

**EOF**, empirical orthogonal function

**GA**, genetic algorithm

**GCE**, Goddard Cumulus Ensemble

**GPU**, Graphics Processing Unit

**L-BFGS**, limited memory Broyden-Fletcher-Goldfarb-Shanno

**ld-dynFWA**, dynamic search fireworks algorithm with linearly decreased dimension number strategy

**MABC**, modified artificial bee colony

**MPI**, Message Passing Interface

**NAO**, North Atlantic Oscillation

**NAOI**, NAO index

**OGE**, optimally growing initial error

**OpenMP**, Open Multi-Processing

**OPR**, optimal precursor

**PCA**, principal component analysis

**PCAFP**, PCA-based flower pollination

**PCAGA**, principal component analysis based genetic algorithm

**PGAPSO**, PCA-based GA and PSO hybrid algorithm

**POP**, Parallel Ocean Program

**PPSO**, PCA-based PSO

**PSO**, particle swarm optimization

**RTM**, River Transport Model

**SIMD**, Single Instruction Multiple Data

**SLP**, sea level pressure

**SPG2**, spectral projected gradient 2

**SQP**, sequential quadratic programming

**WRF**, Weather Research and Forecasting





**XWAV**, Ocean-wave

*Author contributions.* Jing Li wrote the original manuscript and designed the algorithm and parallel scheme; Xiaodan Luo and Jing Li configured the simulation environment and performed the experiments; Bin Mu and Shijin Yuan supported the project, reviewed and edited the manuscript. We also thank the help of Dai as an expert in the area of atmosphere.

*Acknowledgements.* This work was supported by National Natural Science Foundation of China with grant number [41405097]. The calculation of this work was performed on Tianhe-2. Thanks for the support of National Supercomputer Center in Guangzhou (NSCC-GZ).





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
