# Peer review of "Optimal Precursors Identification for North Atlantic Oscillation using CESM and CNOP Method"

_Nonlinear Processes in Geophysics, 2020_

## Referee Comment (RC1) · Anonymous Referee #1 · 17 Aug 2020

In this manuscript, the authors adopted a novel CNOP-based approach to study OPRs of the NAO using CESM. They presented a hybrid intelligence algorithm named PGAPSO to solve CNOP, and applied multiple parallel frameworks to improve efficiency. The effectiveness of the PGAPSO was validated by comparing to the BGM and random method, and the OPRs' structures for both NAO+ and NAO- in two different cases were analyzed. The work is interesting and the algorithm seems to be useful also in other studies of the climate sciences. The revised manuscript has been improved considerably. There are a few points that need to be addressed before I can recommend accepting this manuscript.

1, In this work, the authors studied two cases of different initial conditions. I would suggest the authors to provide more background information about the two cases.

[Figure]

2, The authors explained the Parallelization methods in section 3.3 in great detail, and also presented the corresponding results in section 4.6 with three figures and one table. The contents are good and informative, but since the title of the work has been changed, probably the authors could consider shortening this part, or move some of the contents to supplementary materials?

3, In the Abstract, line 10, what is the last stage of the prediction period? Clearer information should be provided here.

4, On page 3, in line 9, "run" should be "running".

5. On page 3, in line 13, a reference is need here in the end of this sentence.

6. On page 18, in line 19, what is the 25th layer? What is the constrained condition T'2<=100? More detailed information is needed here. For instance, it is better to point out clearly the height of the considered layer, instead of simply using the 25th layer.
* * *

---

## Referee Comment (RC2) · Anonymous Referee #2 · 10 Nov 2020

This manuscript describes the impact of the optimal precursor in modifying the North Atlantic Oscillation (NAO).

In this work, the authors compute the most efficient perturbation in a nonlinear context using a state-of-the-art climate model without the need for an adjoint. In this particular study, the method is applied to ∼weekly NAO changes. This is interesting and show some potential. However, I did not find the paper well written and some of the diagnostics and figures are hard to follow.

I do not recommend this work for publication as it is. With some clarifications and a substantial amount of rewriting, I feel that this work could become publishable.

General Comment:

The main issue of the study is a lack of focus and consistency. After reading the manuscript I am still uncertain on the scientific question(s) tackled here. I can mention three: (1) Perturbation of NAO (i.e., physics), (2) Development of PGAPSO with application to the CESM climate model (i.e., applied mathematics); and (3) Numerical performance of PGAPSO with CESM (numerical science). Of these three question, none was properly answered in the study. This is mostly due to the disparity of the three questions (and their topics) and that the authors tried to tackle them all in a single manuscript.

I would suggest the authors to choose and to focus on a single scientific question and to answer it in details.

Please see my more specific comments below.

Specific Comments:

1) Overall the abstract is not well written (see specific suggestions below). It is quite unclear what is the scientific question, how it will be tackle, and the conclusion of the work.

2) More generally the text is not well written. The English is quite poor and the citation seems to suffer from format problems. More fundamentally there is a lot of terminologies that are not defined (e.g., "NAO event", "Cases") or switch along the manuscript (e.g., CNOP, PGAPSO, OPR). This is not acceptable and extremely confusing.

3) The introduction suffers from the lack of contextualization from other "perturbation" method. For instance it would have been nice to acknowledge other method that has considered change in atmospheric dynamics (SVD). It also suffer from the lack of introduction of fundamental concept mentioned and used in the study. For instance, the predictability of the NAO is not discussed in depth. The concept of predictability is not properly mentioned, whereas it is crucial for the study.

4) The experimental set up does not make sense at all... I do not understand why the

perturbation is restricted to the Polar region (north of 60N). What scientific question can be answered here? I feel that this choice is quite random and does not follow from the vast literature on the topic (or at least it is not motivated that way).

6) Overall there is a lack of consistency and some information are missing making the manuscript hard (if not impossible) to follow. This also highly negatively impacts the reproducibility of the study.

Minor/Technical Comments:

p.1-l.1: replace "seesaw phenomenon" by "variability". I have never seen the term "seesaw" - which implicitly implies an interplay between two regions - associated to the NAO. But I might be wrong.

p1.-l.2: replace "has a profound influence" on by "influences".

p.1-l.1: replace "for" by "of"

p.1-l.3: "NAO event" is not defined yet.

p.1-l.1: replace "the NAO anomaly pattern" by "NAO anomaly"

p.1-l.15: remove "phase reversing" and "in the meridional direction"

p.1-l.16: replace "is mainly" by "can be"

p.1-l.18: replace "mode of atmospheric circulation variability" by "variability mode of the atmospheric circulation"

p.1-l.22: replace "quantified" by "quantitative"

p.1-l.22: replace "difference between normalized SLP" by "normalized difference between SLP"

p.1-l.24: add "over" before "Azores"

p.1-l.24: the term "turbulence" does not make sense here.

p.2-l.2: replace "etc." by "for instance"

p.2-l.2: Please clarify the location of the surface temperature variation mentioned.

p.2-l.3: "The NAO can be regarded as a nonlinear initial value problem" is quite out of context, please clarify or introduce it more specifically.

p.2-l.4: "NAO events" is not define and the association of "NAO" and "event" does not make sense. You define the NAO as variability. What is an event? An extreme value of the variability?

p.2-l.20: The term "gradually" does not make sense.

p.2-l.22: remove the "Jiang et al" in the bracket.

p.2-l.22: remove the "Dai et al" in the bracket.

p.2-l.28: remove the "Marshall and Molteni" in the bracket.

p.2-l.30: remove the "Jiang et al" in the bracket.

p.2-l.30: remove the "Dai et al" in the bracket.

p.2-l.29: the term "optimally growing initial error (OGE)" is hard to follow without further explanation

p.3-l.3-5: This sentence is a strong statement. References are need here.

p.4-l.21: replace "between 60N and 90N" by "north of 60N"

p.5-Fig.2: "The region of the NAO". The NAO is not defined by a region. This does not make any sense.

p.6.-l.1: Please add a reference for "linear singular vector".

p.6-l4-5: The predictability of the NAO was not introduce at all in the intro. The "predictability" concept (and references) neither...

p.6-l.6: "NAO events" is still not defined. At this stage of the manuscript a quantitative definition is needed.

p.6-l.10: It is unusual to write the S_0 inside the bracket. Also, with this notation, it looks like the operator (M, which is a function of S0) is equal to the vector (St).

p.6-l.14: Most term are undefined (S' , \deltaS).

p.6-l16: "NAOI" is not defined.

p.6-l.18-19: "NAOI(NAO+) and NAOI(NAO-)" are not defined and do not make sense.

p.7-l.2: D is not defined

p.7-l.3: 10% of the local variation? Would that not be more usual to take even something lower for predictability such as 1%. Also using a typical variation in time rather than space would be more ideal.

p.7-l.8: Remove "Liu" in the bracket

p.7-Fig.3: "North Atlantic Region [...]" Are you suggesting that your EOF was computed on this restricted region? If so please clarify,

p.8-l.18: Use a ' in the right hand side of the equation, if it is a new variable (i.e., anomaly).

p.9-l.2: Be more quantitative on how smaller the space is.

p.15-Fig.8: How is it built? Is it a composite of NAO+ and NAO- from the 50 iterations of the distribution depicted in Fig.7? Please clarify?

p.15-l.9: I don't understand how the random procedure work? Is it a composite of random perturbation leading to + and - NAO values? If so how many random perturbation were used to build the composite? If the composite is built as a mean of an ensemble, the figure should show the standard deviation to depict the level of uncertainty?

p.15-l.12-13: I did not find the definition of the cases?

p.16-Fig.9: The title should go on top, the x-axis should be labelled.

p.16-l.11-12: For CNOP_NAO+ and CNOP_NAO-, I guess? Please clarify.

p.17-Fig.10: If I understand correctly the random method (i.e., composite/mean of - and + outcomes) the pattern should be symmetric by definition, isn't it? Here it is definitely not! Is it a mistake? Please clarify? Also, replace PGAPSO by CNOP for consistency (and be consistent in all figures and all along the text).

p.18-Fig.12: Add "for the CNOP_NAO+ and CNOP_NAO-" at the end of the caption

p.18-l.17-19: I don't understand that...

p.19-Fig.13: Replace "OPRs" by "CNOPs" in the caption, because you computed other types of "OPRs" (i.e., random and BV). Also be consistent throughout the manuscript (figure and text).

p.20-Fig.15: Which case this figure refer to? Is it the response (on temperature?) of temperature only perturbation? I don't understand the point here... There is a huge lack of information here (and elsewhere), making the reproducibility impossible and making the manuscript extremely hard to follow (if not impossible).

p.23-l24-26: Improving the CESM computation is not without interest, but I don't understand how it fits with the particular target here: CNOP. It would have been more interesting to show how the computation of CNOP can be improved for "constant-performance" of CESM, isn't it? I may be missing the point, but it does not seem align with the rest of the analysis.

---

## Author Comment (AC1) · 18 Dec 2020

Dear reviewer:

We are very grateful for your comments about our manuscript. On behalf of my co-authors, we thank you very much for giving us an opportunity to revise our manuscript. Based on your comments, we have made the following modifications to this manuscript:

1. Comment:

In this work, the authors studied two cases of different initial conditions. I would suggest the authors to provide more background information about the two cases.

Response:

Thank you for your suggestion. The second case was added according to the previous reviewer in order to prove that the CNOP method is effective for different initial conditions. Our experiment is based on the model data and starts simulation from the 53rd winter of the model year (0053-11-01). We select two cases that would not develop into the strong NAO event after 15 days (since the e-folding time scale of the NAO event is two weeks) to observe the NAOI variation after superimposing perturbations. Case 1 is from 0053-01-11 to 0053-01-26, while Case 2 is from 0053-01-15 to 0053-01-30. The reference states of Case1 and Case 2 have the opposite phases, with the NAOI of 0.82 and -1.09. From Figure 9 we can see that the corresponding events can be triggered by CNOP_PO and CNOP_NE, with the NAOI of 3.06 and -3.09 for Case 1 and 2.22 and -2.58 for Case 2. It is indicated that the CNOPs searched by our method can cause the maximum uncertainty under various initial conditions.

Modification:

We add some descriptions of these two cases, especially for why choosing these two cases to simulation the NAO. (See M1)

2. Comment:

The authors explained the Parallelelization methods in section 3.3 in great detail, and also presented the corresponding results in section 4.6 with three figures and one table. The contents are good and informative, but since the title of the work has been changed, probably the authors could consider shortening this part, or move some of the contents to supplementary materials?

Response:

The performance enhancement of the algorithm and numerical model is an innovation point of our research, but it may indeed take up too much space to introduce the parallelization technique. We merged three subsections in Section 'Parallelization' and shorten the contents, in particular for the related works about GPU acceleration.

Modification:

See M2

3. Comment:

In the Abstract, line 10, what is the last stage of the prediction period? Clearer information should be provided here.

Response:

The last stage refers to the period from Day 7 or Day 9 to Day 15. From Figure 8, it can be seen that there has not been much change in NAOI when the perturbations just superposing on the basic state. The curves of perturbations (CNOPs, Breds, and Randoms) start to evolve in different directions on Day 7 (Case 1) or Day 9 (Case 2). It proves that the nonlinear process begins to take effect in the last few days. To make the sentence more clear, we replace the "last stage" with more detailed information here.

Modification:

See M3

4. Comment:

On page 3, in line 9, "run" should be "running".

Response:

Sorry for the incorrect writing. We have modified this word.

Modification:

See M4

5. Comment:

On page 3, in line 13, a reference is need here in the end of this sentence.

Response:

Thank you for pointing this out. We have cited a representative work at the end of this sentence.

Modification:

See M5

6. Comment:

On page 18, in line 19, what is the 25th layer? What is the constrained condition T'2 <= 100? More detailed information is needed here. For instance, it is better to point out clearly the height of the considered layer, instead of simply using the 25th layer.

Response:

CESM uses the hybrid level at midpoints $(1000 * (A + B))$ to express the levels of layers. Assume that hybrid level definitions is: $p = a * p0 + b * ps$, the midpoints level is $p(k) = hyam(k) * ps0 + hybm(k) * ps$. The 25th layer corresponds to the level of 7.389, and it may easy to understand or follow to mark it "25th layer". Relevant information has been added.

We're very sorry that there are some mistakes in the constrained condition. The constrained condition should be $1/D \int_{D} \int_0^1 T^2 d\sigma dD <= 100$ (The summation of the temperature square in the grid of north of 60N should no more than 100).

Modification:

see M6

We appreciate editors and reviewers' warm work earnestly and hope the correction will meet with approval. Once again, thank you very much for your comments and suggestions. If you have any questions, please contact us without hesitation.

Best regards,

Jing Li

---

## Author Comment (AC2) · 18 Dec 2020

Dear reviewer:

Thank you for your comments concerning our manuscript entitled "Optimal Precursors Identification for North Atlantic Oscillation using CESM and CNOP Method". Those comments are all valuable and very helpful for revising and improving our paper. We have studied comments carefully and have made modifications. The response to the comments are as follows:

1. Comment:

The main issue of the study is a lack of focus and consistency. After reading the manuscript. I am still uncertain on the scientific question(s) tackled here. I can mention three: (1) Perturbation of NAO (i.e., physics), (2) Development of PGAPSO with application to the CESM climate model (i.e. applied mathematics); and (3) Numerical performance of PGAPSO with CESM (numerical science). Of these three question, none was properly answered in the study. This is mostly due to the display of the three questions (and their topics) and that the authors tried to tackle them all in a single manuscript.

Response:

Thank you for your comments. I would like to explain the scientific problem studied in this paper. In this paper, we adopt a performance-optimized adjoint-free method called PGAPSO to identify the optimal precursors (OPR) of the North Atlantic Oscillation (NAO) and apply it to the Community Earth System Model (CESM) for which no adjoint model has been developed yet. Using the PGAPSO, we successfully identify the OPRs that can cause strong NAO events and explore the pattern structure of the OPRs along with the variation trends of the NAOI during the simulation period. Moreover, performance optimization using multiple parallel frameworks improves the efficiency of the algorithm, which is also crucial for the research of the problem with such high dimensions. The lengthy description of the PGAPSO approach in this paper is due to the fact that CESM has no adjoint model and the corresponding resolution has a large data scale. The situation is completely different from the previous works of OPR. We try to illustrate how this approach can avoid the use of gradient information and how it can be used for fast solutions with a large data scale and limited computing resources. Therefore, the subject of our manuscript consists of these three parts.

2. Comment:

Overall the abstract is not well written (see specific suggestions below). It is quite unclear what is the scientific question, how it will be tabkle, and the conclusion of the work.

Response:

Thank you for pointing this out. The abstract has been rewritten, and some details about the experiment are added. For instance, the settings of the experiment (variable, region, objective function ..., etc.) and the conclusion. The subject of this paper is to present an application of the adjoint-free CNOP method on the identification of OPRs of the NAO.

Modification:

See M1

3. Comment:

More generally the text is not well written. The English is quite poor and the citation seems to suffer from format problems. More fundamentally there is a lot of terminologies that are not defined (e.g., "NAO event", "Cases") or switch along the manuscript (e.g., CNOP, PGAPSO, OPR). This is not acceptable and extremely confusing.

Response:

Thank you for your suggestions. We have tried our best to modify on writing and expressions for four rounds (The previous version: https://npg.copernicus.org/preprints/npg-2019-25/). According to the previous reviewers' comments, we also ask the artificial proofreading services of Grammarly for help. The writing has been further improved in this round.

We typeset the manuscript using the Latex template, and it should meet the standards of the journal.

The NAO event refers to the state whose NAOI is greater than 1.0 (or less than -1.0) standard deviation for three or more consecutive days in strict definition [1]. In this work, we adopt the simplistic definition, which assumes that the NAO events occur when NAOI > 1.0 or NAOI < -1.0 [2]. We add it to the end of Section 3.1. The case denotes the experimental subject with specific experimental conditions, including initial conditions, simulation period, start date, model parameter, ..., etc.

The definition of these abbreviations can also be found in Appendix A. CNOP is an abstract method to solve predictability problems, such as solving OPRs, solving the optimally growing initial error (OGE), model parameter sensitivity, and so on. The PGAPSO is a kind of specific algorithm to implement the CNOP method, while OPR is a kind of perturbation that would cause the largest uncertainty in the climate simulation.

4. Comment:

The introduction suffers from the lack of contextualization from other "perturbation" method. For instance it would have been nice to acknowledge other method that has considered change in atmospheric dynamics (SVD). It also suffer from the lack of introduction of fundamental concept mentioned and used in the study. For instance, the predictability of the NAO is not discussed in depth. The concept of predictability is not properly mentioned, whereas it is crucial for the study.

Response:

It is important to note that SVD is not a method of generating perturbations. SVD is a feature extraction method that is similar to PCA, which is adopted in our manuscript. Besides, it is hardly necessary to compare PCA with SVD since they follow similar principles. In the introduction, we have presented another work on OPRs of the NAO and introduced the SPG2 algorithm using in their studies. In addition, we have also mentioned other perturbation method along with their characteristics, like SQP, L-BFGS, PCAGA, MABC, PCAFP, ..., etc. The benefits and drawbacks of these approaches are also discussed in the introduction. As for the predictability, we summed up it in a sentence: we aim to find the initial perturbations that cause the largest prediction error under a specific constraint condition at the prediction time. We can avoid the initial error that has a similar spatial structure with OPR, and it is also helpful for determining sensitive areas and intensive observations.

5. Comment:

The experimental set up does not make sense at all. . . I do not understand why the perturbation is restricted to the Polar region (north of 60N). What scientific question can be answered here? I feel that this choice is quite random and does not follow from the vast literature on the topic (or at least it is not motivated that way).

Response:

The region was not selected randomly. We explain in the abstract that both the variables and region are chosen in the sensitivity experiment, which has been performed as the preparatory work of this research. In these experiments, we found that the perturbation in the Polar region has a larger impact on the NAO, thus we aim to explore what extent do the NAO events would be caused by the perturbations in the Polar region. The details of the sensitivity experiment are omitted and only briefly mentioned in the abstract since it is not the emphasis of this paper. The predictability research of the NAO using the CNOP method is still in a blank, and the experience for settings are all from the results of pretests.

6. Minor/Technical Comments:

- p.1-l.1: replace "seesaw phenomenon" by "variability". I have never seen the term "seesaw" - which implicitly implies an interplay between two regions - associated to the NAO. But I might be wrong.

Response: "seesaw" is a commonly used term in the field of geoscience. It refers to the inverse relationship with some specific physical variable between the two regions. It appeared in plenty of related works, for instance, "In the North Atlantic sector, the interaction between the North Atlantic Oscillation (NAO) and a SIC seesaw between the Labrador Sea and Greenland–Barents Sea dominates." [3], "The presence of a low- to mid-latitude interhemispheric hydrologic seesaw is apparent over orbital and glacial-interglacial timescales" [4], "and a hemispheric-scale seesaw-like pattern dominant in sea-ice variability over the wintertime Northern Hemisphere" [5], and so on.

Modification: See M2 - p1.-l.2: replace "has a profound influence" on by "influences". (Modification: see M3)

- p.1-l.1: replace "for" by "of" (Modification: see M4)

- p.1-l.3: "NAO event" is not defined yet. (Modification: see M5)

- p.1-l.1: replace "the NAO anomaly pattern" by "NAO anomaly" (Modification: see M6)

- p.1-l.15: remove "phase reversing" and "in the meridional direction" (Modification: see M7)

- p.1-l.16: replace "is mainly" by "can be" (Modification: see M8)

- p.1-l.18: replace "mode of atmospheric circulation variability" by "variability mode of the atmospheric circulation" (Modification: see M9)

- p.1-l.22: replace "quantified" by "quantitative" (Modification: see M10)

- p.1-l.22: replace "difference between normalized SLP" by "normalized difference between SLP" (Modification: see M11)

- p.1-l.24: add "over" before "Azores" (Modification: see M12)

- p.1-l.24: the term "turbulence" does not make sense here.

Response: The word "turbulence" has been replaced by "fluctuation". Modification: see M13

- p.2-l.2: replace "etc." by "for instance" (Modification: see M14)

- p.2-l.2: Please clarify the location of the surface temperature variation mentioned. (Modification: see M15)

- p.2-l.3: "The NAO can be regarded as a nonlinear initial value problem" is quite out of context, please clarify or introduce it more specifically.

Response: The idea of this manuscript was inspired by the study of Zhina Jiang et al.

[6]. It was mentioned in the abstract that the CNOP method is based on a viewpoint that the NAO is a nonlinear initial-value problem. ("The conditional nonlinear optimal perturbation (CNOP) method is used to explore the optimal precursors that trigger the North Atlantic Oscillation (NAO) anomaly pattern with a triangular T21, three-level, quasi-geostrophic global spectral model based on a viewpoint that the NAO is a nonlinear initial-value problem.") As described in Section "CNOP and PGAPSO", the main idea is converted to the extremum problems related to the initial conditions.

- p.2-l.4: "NAO events" is not define and the association of "NAO" and "event" does not make sense. You define the NAO as variability. What is an event? An extreme value of the variability?

Response: The definition of "NAO events" has been added in the end of Section "CNOP and PGAPSO". Modification: see M5

- p.2-l.20: The term "gradually" does not make sense.

Response: It presents the process from first application to improved versions.

- p.2-l.22: remove the "Jiang et al" in the bracket.

Response: This manuscript was compiled and typeset using the Latex template provided by the journal, and the citation format is also assigned in the template.

- p.2-l.22: remove the "Dai et al" in the bracket.

Response: Same as above.

- p.2-l.28: remove the "Marshall and Molteni" in the bracket.

Response: Same as above.

- p.2-l.30: remove the "Jiang et al" in the bracket.

Response: Same as above.

- p.2-l.30: remove the "Dai et al" in the bracket.

Response: Same as above.

- p.2-l.29: the term "optimally growing initial error (OGE)" is hard to follow without further explanation

Response: The initial error, which causes the largest prediction error under a given constraint, is denoted as the optimally growing initial error (OGE) [7]. Similar to the OPR, the OGE is also a kind of predictability problem to measure the uncertainty in climate predictions. We did not perform experiments of the OGE, thus it is not the focus of our manuscript.

- p.3-l.3-5: This sentence is a strong statement. References are need here.

Response: It was concluded from the paper of Zheng Qin et al published in Nonlinear Processes in Geophysics in 2016 [8]. It has been cited at the end of the sentence.

Modification: see M16

- p.4-l.21: replace "between 60N and 90N" by "north of 60N" (Modification: see M17)

- p.5-Fig.2: "The region of the NAO". The NAO is not defined by a region. This does not make any sense.

Response: Thank you for your suggestion. The caption has been replaced by "The region of perturbations and the North Atlantic sector where the NAO events mainly occur". Modification: see M18

- p.6.-l.1: Please add a reference for "linear singular vector".

Response: The concept of the linear singular vector (LSV) was first proposed by Lorenz, and the reference is added to this sentence [9]. Modification: see M19

- p.6-l4-5: The predictability of the NAO was not introduce at all in the intro. The "predictability" concept (and references) neither...

Response: In the field of atmospheric sciences, there are two kinds of predictability

problems: the model sensitivity to inaccurate initial conditions (first kind) and inaccurate boundary conditions (second kind). The research of OPR can be viewed as the discussion for the first kind of predictability problem. Relevant descriptions are added in the introduction. Modification: see M20

- p.6-l.6: "NAO events" is still not defined. At this stage of the manuscript a quantitative definition is needed. (Modification: see M5)

- p.6-l.10: It is unusual to write the $S_0$ inside the bracket. Also, with this notation, it looks like the operator (M, which is a function of S0) is equal to the vector (St).

Response: Indeed, M is the nonlinear operator, and $S_0$ (basic state) is the variable. It means that $S_0$ is passed by the operator M from $t_0$ (initial time) to t, and $S_t$ is obtained.

- p.6-l.14: Most term are undefined (S' , \deltaS).

Response: $S_t'$ is the final state generated by superimposing perturbation $s_0$ on the basic state $S_0$, and $S_t$ is the reference state without perturbations. \delta S is the difference between these two final states $S_t'$ and $S_t$.

- p.6-l16: "NAOI" is not defined.

Response: NAOI was defined in the introduction (p.1-l.24) "The NAO index (NAOI) is a quantified indicator of the NAO, and its classical definition is the difference between normalized SLP over Iceland and Azores (Andersson, 2002)". The equation of the NAOI is presented in equation (7).

- p.6-l.18-19: "NAOI(NAO+) and NAOI(NAO-)" are not defined and do not make sense.

Response: NAO^+ refers to the phase with positive NAOI, while NAO^+ refers to the phase with a negative NAOI. It is the common sense of the research of the NAO. Since the NAO has two types of phases (NAO^+ and NAO^-), we need to search the $s_0^*$ in two directions, which correspond to the maximum and minimum of the function J. The

experiments for NAOˆ+ and NAOˆ- are performed separately.

- p.7-l.2: D is not defined

Response: According to equation (6), the constraint condition is the 10% summation of the kinetic energy of the basic state. Therefore, area D is the region of perturbation, which is located north of 60N.

- p.7-l.3: 10% of the local variation? Would that not be more usual to take even something lower for predictability such as 1%. Also using a typical variation in time rather than space would be more ideal.

Response: The constraint condition \epsilon is to limit the perturbation to a reasonable range. 10% of the kinetic energy is proved to be appropriate for this experiment and is a constant for a certain case. Before an iteration, when we need to specify the range of the perturbation, the given conditions only contain the basic state (at the initial time) and the perturbation. At this time, we cannot obtain the state at any time except for the initial time. Therefore, using a typical variation in time is not feasible.

- p.7-l.8: Remove "Liu" in the bracket

Response: The citation format is specified in the Latex template, and we may need to follow the format.

- p.7-Fig.3: "North Atlantic Region [...]" Are you suggesting that your EOF was computed on this restricted region? If so please clarify

Response: The EOF was performed in the North Atlantic region. We have clarified the region where EOF is computed in the caption of Figure 3, "The first mode of the EOF with SLP anomaly field concentrated in the North Atlantic region between 90◦W - 40◦E, 20◦N - 80◦N."

- p.8-l.18: Use a ' in the right hand side of the equation, if it is a new variable (i.e., anomaly).

Response: Thank you for your suggestion, and the P_ij has been replaced by P_ij'.

Modification: see M21

- p.9-l.2: Be more quantitative on how smaller the space is.

Response: The number of dimensions of the original space is 586,1376, and the reduced dimension size is 50. The search domain scale has been shrunk by about 99.99915%. It has been added at the end of the sentence.

Modification: see M22

- p.15-Fig.8: How is it built? Is it a composite of NAO+ and NAO- from the 50 iterations of the distribution depicted in Fig.7? Please clarify?

Response: Yes, we conduct 50-iteration experiments on different simulation duration and obtain Figure 8.

Modification: see M23

- p.15-l.9: I don't understand how the random procedure work? Is it a composite of random perturbation leading to + and - NAO values? If so how many random perturbation were used to build the composite? If the composite is built as a mean of an ensemble, the figure should show the standard deviation to depict the level of uncertainty?

Response: I'm afraid you misunderstand the process of the random method. Taking the PGAPSO as an example, the direction and speed of perturbation are determined by the update formula. In the random method, the perturbations are generated randomly in each iteration, with random position and random speed (in a reasonable range). Each perturbation is independent to compute the fitness value, and no composite was built. The number of perturbations in each iteration is the same for these two methods to ensure fairness.

- p.15-l.12-13: I did not find the definition of the cases?

Response: Both of the cases are the 15-day periods in the 53rd model year, with normal NAOI and different initial conditions. These two cases selected in this manuscript do not have specificity and are only briefly introduced in Section 4.4.

Modification: see M24

- p.16-Fig.9: The title should go on top, the x-axis should be labelled.

Response: The titles are removed to the top, and the label of the x-axis is "Days", which is on the left of the x-axis.

Modification: see M25

- p.16-l.11-12: For CNOP_NAO+ and CNOP_NAO-, I guess? Please clarify.

Response: Thank you for your suggestion. The relevant information has been added to this part.

Modification: see M26

- p.17-Fig.10: If I understand correctly the random method (i.e., composite/mean of - and + outcomes) the pattern should be symmetric by definition, isn't it? Here it is definitely not! Is it a mistake? Please clarify? Also, replace PGAPSO by CNOP for consistency (and be consistent in all figures and all along the text).

Response: First, the random method does not acquire the composite or mean of the outcomes. It selects the perturbation with the largest (smallest) fitness value from a large number of perturbations generated randomly. Such an extreme value problem in a massive-scale space is an NP-hard problem, and we are scarcely possible to find the unique right solution. And further, since the NAO is a nonlinear process ($M_{\{t_0 \to t\}}$), and its evolution direction is affected greatly by the initial condition. The pattern of positive phase ($NAO\hat{}+$) and negative phase ($NAO\hat{}-$) would not be symmetric even if we obtain the perturbations that can cause the largest uncertainty. For another question, CNOP is an abstract method that studies the OPR by generating and superimposing

perturbations on the initial field. Multiple algorithms can be adopted to implement it, such as SPG2, gradient definition, and intelligent algorithms, like the PGAPSO. In this part, the PGAPSO and the random method are approaches that implement the CNOP method to find the OPRs. Therefore, it is not appropriate to replace PGAPSO with CNOP.

- p.18-Fig.12: Add "for the CNOP_NAO+ and CNOP_NAO-" at the end of the caption

Modification: see M27

- p.18-l.17-19: I don't understand that. . .

Response: We discuss the multi-variable perturbations, which contains zonal wind, meridional wind, temperature, specific humidity, surface humidity, surface pressure, and surface geopotential. We find that the perturbation only includes temperature can also cause the NAO event, then we generate the perturbation with only one variable and conduct the following experiments. The results of the experiments shown that temperature has an important impact on the NAO evolution. - p.19-Fig.13: Replace "OPRs" by "CNOPs" in the caption, because you computed other types of "OPRs" (i.e., random and BV). Also be consistent throughout the manuscript (figure and text). (Modification: see M28)

- p.20-Fig.15: Which case this figure refer to? Is it the response (on temperature?) of temperature only perturbation? I don't understand the point here... There is a huge lack of information here (and elsewhere), making the reproducibility impossible and making the manuscript extremely hard to follow (if not impossible).

Response: Thank you for pointing it out. The experiment is conducted based on the initial condition of Case 1. The dimension number of the reduced space and iteration number are the same as the previous experiment. We correct a mistake in the constrained condition here, and the formula should be $1/D \int_{D} \int_0^1 T^2 'd\sigma d D <= 100$ (The summation of the temperature square in the grid of north of 60N).

Modification: see M29

- p.23-l24-26: Improving the CESM computation is not without interest, but I don't understand how it fits with the particular target here: CNOP. It would have been more interesting to show how the computation of CNOP can be improved for "constant-performance" of CESM, isn't it? I may be missing the point, but it does not seem align with the rest of the analysis.

Response: In this manuscript, the performance optimization of solving CNOPs is conducted from two perspectives: the parallelization of CESM and the acceleration of the PGAPSO. Since the efficient bottleneck of the entire process mainly focuses on model integration, we try to reduce the latency time as much as possible. The calculation of fitness value is fast enough and little need for enhancement. Thus, the computation process of CNOP has been optimized by MPI, which is shown in Figure 4. Each particle is assigned an independent process and can calculate the fitness value concurrently. Combining these two means of optimization, the computation of CNOP achieves high computational efficiency.

References:

[1] Dai, Guokun , M. Mu , and Z. Jiang . "Evaluation of the Forecast Performance for North Atlantic Oscillation Onset." ADVANCES IN ATMOSPHERIC SCIENCES (2019).

[2] Dai, Guokun , M. Mu , and Z. Jiang . "Relationships between Optimal Precursors Triggering NAO Onset and Optimally Growing Initial Errors during NAO Prediction." Journal of the Atmosphericences 73.1(2015):293-317.

[3] Frankignoul, Claude, et al. "Observed Atmospheric Response to Cold Season Sea Ice Variability in the Arctic." Journal of Climate, vol. 27, no. 3, 2014, pp. 1243–1254.

[4] Lechleitner, Franziska A., et al. "Tropical Rainfall over the Last Two Millennia: Evidence for a Low-Latitude Hydrologic Seesaw." Scientific Reports, vol. 7, no. 1, 2017, p. 45809.

[5] Yamamoto, Kentaro, et al. "Intra‐seasonal Relationship between the Northern Hemisphere Sea Ice Variability and the North Atlantic Oscillation." Geophysical Research Letters, vol. 33, no. 14, 2006.

[6] Jiang, Zhina, et al. "A Study of the North Atlantic Oscillation Using Conditional Nonlinear Optimal Perturbation." Journal of the Atmospheric Sciences, vol. 70, no. 3, 2013, pp. 855–875.

[7] Mu, Mu, et al. "Similarities between Optimal Precursors for ENSO Events and Optimally Growing Initial Errors in El Niño Predictions." Theoretical and Applied Climatology, vol. 115, no. 3, 2014, pp. 461–469.

[8] Zheng, Qin, et al. "Conditional Nonlinear Optimal Perturbations Based on the Particle Swarm Optimization and Their Applications to the Predictability Problems." Nonlinear Processes in Geophysics, vol. 24, no. 1, 2016, pp. 101–112.

[9] Lorenz, Edward N. . "A study of the predictability of a 28-variable atmospheric model." Tellus 17.3(1965):321–333.

We appreciate your comments, and these comments are very helpful. We have made great changes according to these comments. The revised portions are marked with orange boxes and are noted with labels. We are very sorry for our incorrect writing and confused expression. We tried our best to revise the manuscript. If you have any questions, please contact us without hesitation. Thank you very much for all your help, and looking forward to hearing from you soon.

Best regards,

Jing Li